# Global and regional radiative forcing from 20% reductions in BC, OC and SO4 - an HTAP2 multi-model study

Camilla Weum Stjern[1], Bjørn Hallvard Samset[1], Gunnar Myhre[1], Huisheng Bian[2], Mian Chin[3], Yanko Davila[4], Frank Dentener[5], Louisa Emmons[6], Johannes Flemming[8], Amund Søvde Haslerud[1], Daven Henze[4], Jan Eiof Jonson[7], Tom Kucsera[9], Marianne Tronstad Lund[1], Michael Schulz[7], Kengo Sudo[10], Toshihiko Takemura[11], Simone Tilmes[6]

[1] CICERO Center for International Climate and Environmental Research, Oslo, Norway
[2] Goddard Earth Sciences and Technology Center, University of Maryland, Baltimore, Maryland, USA
[3] Earth Sciences Division, NASA Goddard Space Flight Center, Greenbelt, MD, USA
[4] Department of Mechanical Engineering, University of Colorado, Boulder, CO, USA
[5] European Commission, Joint Research Centre, Institute for Environment and Sustainability, Ispra (VA), Italy
[6] Atmospheric Chemistry Division, National Center for Atmospheric Research (NCAR), CO, USA
[7] Norwegian Meteorological Institute, Oslo, Norway
[8] European Centre for Medium Range Weather Forecast (ECMWF), Reading, UK
[9] Universities Space Research Association, Greenbelt, MD, USA
[10] Nagoya University, Furocho, Chigusa-ku, Nagoya, Japan
[11] Research Institute for Applied Mechanics, Kyushu University, Fukuoka, Japan

*Correspondence to:* Camilla W. Stjern (camilla.stjern@cicero.oslo.no)

## Abstract

In the Hemispheric Transport of Air Pollution Phase 2 (HTAP) exercise, a range of global atmospheric general circulation and chemical transport models performed coordinated perturbation experiments with 20 % reductions in emissions of anthropogenic aerosols, or aerosol precursors, in a number of source regions. Here, we compare the resulting changes in the atmospheric load and vertically resolved profiles of black carbon (BC), organic aerosols (OA) and sulfate ($SO_4$) from 10 models that include treatment of aerosols. We use a set of temporally, horizontally and vertically resolved profiles of aerosol forcing efficiency (AFE) to estimate the impact of emission changes in six major source regions on global radiative forcing (RF) pertaining to the direct aerosol effect, finding values between. 51.9 and 210.8 $mWm^{-2}$ $Tg^{-1}$ for BC, between -2.4 and -17.9 $mWm^{-2}$ $Tg^{-1}$ for OA, and between -3.6 and -10.3 $Wm^{-2}$ $Tg^{-1}$ for $SO_4$. In most cases, the local influence dominates, but results show that mitigations in South and East Asia have substantial impacts on the radiative budget in all investigated receptor regions, especially for BC. In Russia and the Middle East, more than 80 % of the forcing for BC and OA is due to extra-regional emission reductions. Similarly, for North America, BC emissions control in East Asia is found to be more important than domestic mitigations, which is consistent with previous findings. Comparing fully resolved RF calculations to RF estimates based on vertically averaged AFE profiles allows us to quantify the importance of vertical resolution to RF estimates. We find that locally in the source regions, a 20 % emission reduction strengthens the radiative forcing associated with $SO_4$ by 25 % when including the vertical dimension, as the AFE for $SO_4$ is strongest near the surface. Conversely, the local RF from BC weakens by 37 % since BC AFE is low close to the ground. The fraction of BC direct effect forcing attributable to inter-continental transport, on the

other hand, is enhanced by one third when accounting for the vertical aspect, because long-range
transport primarily leads to aerosol changes at high altitudes, where the BC AFE is strong. While the
surface temperature response may vary with the altitude of aerosol change, the analysis in the present
study is not extended to estimates of temperature or precipitation changes.

## 1. Introduction

Atmospheric aerosols have a range of effects on the atmosphere, biosphere and on human beings.
They significantly alter the global radiative balance, through processes spanning from direct
interaction with sunlight (Myhre et al., 2013; Yu et al., 2006) to modification of cloud properties
(Lohmann and Feichter, 2005; Stevens and Feingold, 2009) and influences on thermal stability (Koch
and Del Genio, 2010). Aerosols have also been shown to affect regional precipitation (Liu et al., 2011;
Khain, 2009) and atmospheric circulation patterns (Bollasina et al., 2011). In addition to climatic
impacts come the adverse effects that aerosol pollution has on human health (Janssen, 2012; Geng et
al., 2013). Changes in aerosol emissions are therefore of interest both for climate and public health
policies (Shindell et al., 2012), which makes it imperative to provide precise estimates of aerosol
effects on these outcomes. However, present day emissions have high spatial and temporal variability,
and acquiring accurate measurements is a challenge. Similarly, aerosol atmospheric lifetimes and
processes leading to long-range transport are insufficiently quantified. The total anthropogenic aerosol
radiative forcing (RF) since the onset of the industrial period counters large parts of the positive RF
from $CO_2$ and other greenhouse gases, and was recently evaluated to be $-0.9$ W m$^{-2}$ with a 95 %
uncertainty interval from $-1.9$ to $-0.1$ W m$^{-2}$ (Boucher et al., 2013). Of the total aerosol RF, the direct
short-wave aerosol radiative interaction contributed with $-0.35$ W m$^{-2}$, with an uncertainty interval of
$-0.85$ to $+0.15$ W m$^{-2}$. These large uncertainty intervals imply that the RF from aerosols is poorly
constrained. Likewise, there is still a large divergence between model- and satellite-derived surface
particulate matter and observed concentrations (Brauer et al., 2016).
One specific uncertainty in calculating aerosol RF is connected to the vertical distribution of aerosols.
The radiative impact of an aerosol depends on its absorbing and reflecting properties, but these
properties, as well as their radiative impact, are subject to modifications by variable atmospheric
conditions. For instance, relative humidity has a large impact on the scattering properties of light
reflecting aerosols (Fierz-Schmidhauser et al., 2010; Haywood and Shine, 1997). Also, the radiative
forcing efficiency of absorbing aerosols is augmented with increasing quantities of underlying clouds
and gases that reflect solar radiation back onto the aerosols, thereby enhancing their absorption
(Zarzycki and Bond, 2010). Meanwhile, competition with other processes such as Rayleigh scattering
and radiative interactions of other aerosol species (Samset and Myhre, 2011) may dampen the
radiative impact of an aerosol. As these factors typically vary with altitude, so will the aerosols'
forcing efficiency. Accurate knowledge of the vertical distribution of aerosol load is therefore
important (Ban-Weiss et al., 2012; Samset and Myhre, 2015; Vuolo et al., 2014; Zarzycki and Bond,
2010). Presently, the atmospheric models that simulate the climate impact of aerosols have substantial
variations in their vertical distribution of aerosols. In fact, results from the recent AeroCom Phase II
multimodel exercise (Samset et al., 2014; Samset et al., 2013) show that differences in vertical profiles
gave rise to between 20 % and 50 % of the intermodel differences in direct RF estimated from
common BC emissions from fossil fuel and biofuels (FF+BF).
Due to long-range atmospheric transport, emissions in major source regions may have widespread
health and climate impacts that go far beyond the domestic domain. Studies of long-range transport of
aerosols have found that the vertical distribution of aerosols in the source region has important
implications to the magnitude and spatial extent of their climate impact – not only because of the
variation of forcing efficiency with height, but because the strong large-scale winds in the upper
troposphere can transport aerosols for particularly long distances if they reach these levels. For
instance, Liu et al. (2008) found in a study of Cloud-Aerosol Lidar and Infrared Pathfinder Satellite
Observations (CALIPSO) measurements that the higher Saharan dust aerosols were lifted up in the
source region, the further they were carried across the Atlantic Ocean. Similarly, Huang et al. (2008)
studied long-range transport from Asia during the  Pacific Dust Experiment (PACDEX) and found
indications of aerosol transport via upper tropospheric westerly jets – the efficiency of which was
influenced by the vertical distribution of Asian dust in the free troposphere of the source region.
These studies underline the need for a better understanding of how variations between atmospheric
models contribute to the uncertainties in radiative forcing estimates, and specifically the role of
different vertical distribution of aerosols to these uncertainties. In 2005, the Task Force on
Hemispheric Transport of Air Pollution (TF HTAP) was established under the United Nations
Economic Commission for Europe (UNECE) Convention on Long- Range Transboundary Air
Pollution (LRTAP Convention). One of its goals is to further our understanding of aerosol
intercontinental transport, and assess impacts of emission changes on air quality, climate, and
ecosystems (http://www.htap.org/). The climate impact of aerosol emission reductions in four large
source regions was investigated for a series of model simulations from the first phase of the HTAP
Task Force (HTAP1) by Yu et al. (2013), who calculated radiative forcing as the product between
aerosol optical depth and an aerosol forcing efficiency (AFE) estimated using the Goddard Chemistry
Aerosol Radiation and Transport (GOCART) model. They found that when all anthropogenic
emissions where reduced by 20 % in North America, Europe, South Asia or East Asia, the four-region
average global direct radiative forcing of $SO_4$, particulate organic matter and black carbon was
lowered about 9 %, 3 % and 10 %, respectively. Together, the four-region total emissions accounted
for 72 %, 21 % and 46 % of global emissions for $SO_4$, particulate organic matter and black carbon,
respectively. Inter-model differences were found to be substantial, in part because the models were
using different emission inventories in their simulations.
The present study utilizes model experiments organized by the second phase of the TF HTAP
(HTAP2). We focus on the six priority source regions (Fig. 1) selected by the TF HTAP for HTAP2:
North America (NAM), Europe (EUR), South Asia (SAS), East Asia (EAS),
Russia/Belarussia/Ukraine (RUS) and the Middle East (MDE). Note that while the first four regions
are similar to those investigated by Yu et al. (2013), the HTAP2 regions are defined by geopolitical
boundaries while the HTAP1 regions were larger and included more ocean areas. We aim to explain
how much a 20 % emission reduction in these source regions would impact other regions in terms of
aerosol burden and radiative forcing changes. To estimate the climate impacts of the mitigations we
calculate radiative forcing based on column averaged aerosol fields and AFE estimates in a method
equivalent to Yu et al. (2013) (here, using the OsloCTM2 model), but we extend the analyses to also
involve 4D AFE and aerosol burden profiles. This allows us to quantify how the vertical distribution
of aerosols influences the potential impact of regional emission mitigation strategies.
Previous studies have shown that the relationship between instantaneous BC RF, which is what we
estimate here, and the resulting surface temperature change also depends on the altitude of the BC.
The dependence is however not the same as for instantaneous RF. Although not found in all studies
(Ming et al., 2010), indications are that near the surface BC causes strong warming, through the
middle of the troposphere it is only weakly warming, whereas near the tropopause and in the
stratosphere, BC may even cause surface cooling (Ban-Weiss et al., 2012; Samset and Myhre, 2015;
Sand et al., 2013a; Shindell and Faluvegi, 2009). The difference is related both to the indirect and
semi-direct impacts of BC on clouds, both of which cause negative RF, due, respectively, to
microphysical impacts on cloud albedo, and changes in cloud cover due to alterations in atmospheric
stability and relative humidity. It is beyond the scope of this study to calculate climate change in terms
of surface temperature change, and we stress that a positive/negative estimate of direct RF here should
not be translated directly into warming or cooling.
In the next section, we will go through our methods. Section 3 presents the results, starting with
changes in aerosol concentrations for the different experiments, and moving on to resulting changes in
radiative forcing as well as the influence of inter-continental transport. The results are summarized in
Sect. 4.
## 2.  Methods
## 2.1    The HTAP2 experiments and models
As part of the HTAP2 exercise, global aerosol-climate CTMs and GCMs performed a baseline (*BASE*)
simulation with climate and aerosol emissions corresponding to present day (year 2010) conditions
(Galmarini, 2016). Anthropogenic emissions followed Janssens-Maenhout et al. (2015), which for
year 2010 give global BC, OC and $SO_2$ emissions of 5.56, 12.58 and 106.47 Tg species/year,
respectively. Each model also ran simulations with all anthropogenic emissions reduced by 20 % in a
selection of source regions. We have chosen to focus on the six priority source regions pointed out by
the TF HTAP and shown in Fig. 1 (a). The experiments where all anthropogenic emissions are reduced
by 20 % in the NAM, EUR, SAS, EAS, RBU and MDE regions are referred to correspondingly as
*NAMreduced*, *EURreduced*, *SASreduced*, *EASreduced, RBUreduced* and *MDEreduced*. We will
additionally analyze emission reduction influences on the Arctic, also marked in Fig. 1 (a).
The present study takes input from ten global aerosol models, listed in Table 1 along with core
parameters and references. Horizontal and vertical resolutions of the models are also indicated in
Table 1. The time resolution of output used in this study is monthly for all models, although models
were run at finer resolution. To be included here, we required that the models had provided 3D,
temporally resolved mass mixing ratios of atmospheric aerosols for both the baseline and at least four
of the reduced emission scenarios. All models used prescribed meteorology for the year 2010.
Obviously, the use of one specific year will impact the results as prevailing wind patterns and
precipitation levels in the different source regions will vary from year to year, which will influence
transport and removal processes. For instance, 2010 marked the beginning of the strong 2010-2012 La
Niña event, which has been shown to be associated with above-normal intensities of the Asian
monsoon (Goswami and Xavier, 2005).
The analyzed aerosol species include sulfate ($SO_4$), organic aerosols (OA) and black carbon (BC). A
limitation of the current analyses of OA is that while some models reported OA directly, others gave
emissions and concentrations of OC instead (see Table S-1). OC can be converted to OA through
multiplication by an OC-to-OA conversion factor, which varies with source, aerosol age and the
presence of other chemical species (see e.g. Tsigaridis et al. (2014) and references therein). However,
due to limited level of detail in the available model data, as well as due to consistency to the method
used in Chin et al. (manuscript in preparation), we multiplied all OC values by a factor 1.8 to obtain
OA. As some of the models have included secondary organic aerosols (SOA) in their OA values while
other have not, this approximation likely leads to additional inter-model variability.
Model output was provided as mass mixing ratio (MMR, unit of µg/kg), but we have also analyzed the
data in terms of column integrated aerosol abundance. The conversion from MMR to column
abundance was done by interpolating the MMR fields from each model to the resolution of one host
model (OsloCTM2) with a vertical resolution of 60 layers, using pressure and mass of air distributions
from that model and summing over all layers. See e.g. Samset et al. (2013) for a detailed description of
this method.

## 2.2    Estimating radiative forcing

None of the participating models performed native RF calculations. In order to estimate the radiative
forcing resulting from the emission and subsequent concentration reductions simulated by the HTAP2
experiments, we therefore utilized precalculated 4D distributions of aerosol forcing efficiency (AFE),
which is defined as the RF per gram of a given aerosol species. For the three aerosol species, AFE was
calculated for each grid cell and month by inserting a known amount of aerosol within a known
background of aerosols and clouds, for each model layer individually, and calculating the resulting
radiative effect using an 8-stream radiative transfer model (Stamnes et al., 1988) with four short wave
spectral bands (Myhre et al., 2009). I.e. the model was used to calculate the response to a change in
aerosol concentration at a given altitude, and run for a whole year to capture seasonal variability. The
simulations for different model layers were then combined into a set of radiative kernels, one for each
aerosol species. For the radiative transfer calculations aerosol optical properties were derived from
Mie theory. The absorption of aged BC was enhanced by 50% to take into account external mixing, as
suggested by Bond and Bergstrom (2006), and for all models we assume the same mixing ratio
between aged and non-aged BC as in OsloCTM2. Hygroscopic growth of $SO_4$ was included, scaling
with relative humidity according to Fitzgerald (1975). See Myhre et al. (2004) and Myhre et al. (2007)
for a discussion on the impacts of this choice. For OA, purely scattering aerosols are assumed.
Background aerosols were taken from simulations using OsloCTM2.  See Samset and Myhre (2011)
for details, but note that all numbers have been updated since that work, taking into account recent
model improvements (Samset and Myhre, 2015). The resulting AFE profiles, averaged over the
individual regions from Fig. 1 (a), is presented in Sect. 3.3. For a full discussion on the impact on
radiative forcing from using a single model kernel, see Samset et al. (2013). Briefly, multi-model
average forcing becomes representative of that of the host model, including cloud fields and optical
properties, while the variability around this value is indicative of the impact of differences in 3D
aerosol burdens. The resulting reduction in multi-model relative standard deviation depends on the
regional and vertical differences in AFE, but is generally less than 20%.
The direct RF from a given aerosol species due to a 20 % emission reduction was then estimated by
multiplying the modelled aerosol burden change profile $\Delta BD$ (from a given HTAP2 model and
experiment) with the OsloCTM2 AFE distribution for that species and point in space and time (month
of the year):
$$RF(lon, lat, lev, time) = \Delta BD(lon, lat, lev, time) \times AFE(lon, lat, lev, time) \qquad (1)$$
The RF calculated at each model level using this method should be interpreted as the instantaneous
radiative forcing exerted at top of the atmosphere (TOA), due to the aerosol abundance within that
layer.
As mentioned above, using this procedure means that intermodel variability will likely be lower than if
the models had provided their own estimates of RF, and that the absolute RF will be influenced by the

mean efficiency of the host model (OsloCTM2). As recently shown in the AeroCom Phase II model intercomparison (Myhre et al., 2013), OsloCTM2 is among the models with strongest global, annual mean AFE values for BC and OA, in part due to the heightened complexity of the radiation scheme used (Myhre and Samset, 2015). For $SO_4$, the AFE of OsloCTM2 is close to the AeroCom median.

Validation of these kernel estimates against natively calculated RF was not possible in this analysis, as no RF values were available from the model groups. However, Samset et al. (2013) performed a comparison between model simulated and kernel estimated RF and found that for BC, between 20 and 50% of the variability could be attributed to vertical BC profiles alone, with the rest being due to a combination of optical properties, horizontal transport and differences in cloud fields. Also note that Stier et al. (2013) investigated model uncertainty in direct RF for twelve AeroCom models and found substantial diversity in both clear- and all-sky RF even when aerosol radiative properties were prescribed.

As will be shown below, there are significant differences between the vertical profiles of aerosol abundance predicted by the participating models. To estimate the effect of these differences on global, annual mean RF, we also compute the radiative forcing in a way that does not account for the vertical aerosol distributions: we average out the vertical dimension by calculating column aerosol burdens and multiply by corresponding full column AFE distributions from OsloCTM2, which utilized the specific vertical aerosol distribution of that model.

$$RF_{3D}(lon, lat, time) = \Delta BD(lon, lat, time) \times AFE(lon, lat, time) \qquad (2)$$

Here, $RF_{3D}$ indicates a radiative forcing estimate where the two horizontal dimensions, as well as time, is included, but where the vertical dimension is averaged out. For further details on the above method, see Samset et al. (2013).

## 2.3 Response to extra-regional emission reductions

The impact of intercontinental transport between regions is investigated through calculating the Response to Extra-Regional Emission Reductions (RERER). While this metric is originally defined in HTAP (2010) to study the influence of inter-continental transport on region average burden change or surface concentrations, we utilize a version of the RERER defined in HTAP (2010) studying instead the influence on forcing:

$$RERER_{sr} = \frac{\Delta RF_{base,global} - \Delta RF_{base,sr}}{\Delta RF_{base,global}} = \frac{(RF_{base} - RF_{global}) - (RF_{base} - RF_{sr})}{RF_{base} - RF_{global}} \qquad (3)$$

Here, *base* refers to the base simulation with no emission reductions, *global* refers to an experiment where anthropogenic emissions all over the globe are reduced by 20 %, and *sr* refers to the experiment where emissions in source region *sr* are reduced by 20 %. RERER is then calculated for all source regions and species. A low RERER value means that the forcing within a region is not very sensitive to extra-regional emission reductions.

In addition to the above calculation of RERER for RF, we also calculate RERER for changes in total column aerosol burden, which gives an estimate of inter-continental transport in two dimensions (ignoring the vertical).

## 3.  Results and discussion

In the following sections, we first present the global and regional aerosol burdens simulated by the participating models in response to the baseline emissions, before moving on to showing the local and remote burden changes due to 20 % reduction in regional emissions. Then, we show the calculated radiative forcing from these burden changes, and discuss how regional aerosol mitigation efforts may impact local and remote regions.

### 3.1     Baseline aerosol burdens and emissions

Figures 1 (b) – (d) show the multi-model median column integrated burden fields for BC, OA and $SO_4$, respectively, for the unperturbed *BASE* simulation. The source regions of focus in this study are mostly recognized as regions of high aerosol burden in the maps, as are other regions such as Central Africa and South America (high BC and OA from open biomass burning). Areas with significant loads can also be seen over global oceans, far from the main emission regions, showing the importance of long-range aerosol transport for both the global and regional climate impact of aerosols.

In Table 2, the regional averages of aerosol burdens for the four source regions reveal some differences between the regions. Particularly, for BC and OA, East and South Asia have significantly higher burdens than North America, Europe, Russia/Belarussia/Ukraine (henceforth referred to as Russia, for simplicity) and the Middle East. For $SO_4$, the Middle East ranks among the high-emission source regions. The source regions are also different in terms of meteorology (see Table 2) and surface albedo (not shown), which will influence the local as well as remote effects of emission reductions. For instance, the amount, timing and intensity of precipitation events largely controls the rate of wet removal of fresh aerosols. For year 2010 the average daily precipitation in the Middle East was 0.4 mm/day, while in South Asia it was 3.3 mm/day (Table 2). Meanwhile, the South Asian region is also marked by a significantly higher convective mass flux than the other regions, which likely enhances long range transport due to convective lifting of insoluble aerosols to high altitudes. The fractions of BC, OA and $SO_4$ to the total BC+OA+$SO_4$ sum are on the other hand quite similar between the regions, with BC contributing 4-8 % of the total, OA contributing 25-45 % of the total, and $SO_4$ contributing 51-70 % of the total (not shown). Europe has a lower fraction of OA and a higher share of $SO_4$ than the other regions, while the Middle East has a lower BC fraction and higher $SO_4$ fraction.

The relative inter-model standard deviation in emissions is given in the top row of Fig. 2, and demonstrates that for all three species the models disagree the most over the tropics and over the poles. Regionally and annually averaged emissions (second row of Fig. 2) for all three aerosol species are highest in East Asia. The error bars indicate the full range of model results. For BC and $SO_4$ there is a very limited spread between the models, as all HTAP2 model groups used emission data from the Emissions Database for Global Atmospheric Research (EDGAR) HTAP_v2 emission inventory (Janssens-Maenhout et al., 2015). However, there is a large spread in OA emissions between the models, primarily due to high OA emissions from GEOS5, GEOSCHEMADJOINT and GOCART, but presumably also linked to the above mentioned conversion from OC to OA for some of the models, as well as model differences in the treatment of SOA.

In spite of the unified emissions, total aerosol burdens (not shown) vary substantially between the models. This is expected, as there is a broad range of model processes that connect emissions to global aerosol burden, and different models treat these processes differently. For example, the convection schemes used by the different models listed in Table 1 differ markedly. Parametrizations of processes such as wet removal and oxidation will also be sources of inter-model difference, as will their horizontal and vertical resolution. For instance, Molod et al. (2015) performed model simulations of

different horizontal resolution with the GEOS5 model, which parameterizes convection using the relaxed Arakawa-Schubert algorithm (RAS). They found that the mass flux decreases with increasing resolution, resulting in reduced low-level drying, which again might increase wet removal and lower the aerosol burden. Kipling et al. (2015) investigated processes important for the shape of vertical aerosol profiles by performing a number of sensitivity tests using the HadGEM3-UKCA model, and comparing the variation in results to the inter-model variation from the AeroCom Phase II control experiment. They found that the vertical profile was controlled mainly by convective transport, in-cloud scavenging and droplet growth by condensation – processes that have widely different parametrizations between models.

An HTAP2 model-observation comparison study by Chin et al. (manuscript in preparation) finds that in general, compared to measurements, the two CHASER models typically report too high surface concentrations of $SO_4$, OA and BC, while OsloCTM3_v02 generally have low values. Figure 3 shows vertically resolved plots of globally averaged mass mixing rations (MMR) for the three aerosol species, and illustrates that the high values for CHASERT42 and CHASERT106 extend through all vertical layers. It is interesting to note that the CHASER models use a version of the Arakawa-Schubert parametrization of convection, and that the highest-resolution version (T106) has the lowest aerosol burden among the two, which could be related to the findings of Molod et al. (2015) noted above. Note that for $SO_4$, GOCART and GEOS5 have particularly high MMR aloft, see Fig. 3 (c).

## 3.2    Aerosol changes

The third row of Fig. 2 shows the change in global, annual mean aerosol burden following a 20 % emission reduction in the region noted on the x axis. The burden change is clearly highest for the regions with the highest baseline emissions (second row of Fig. 2). The ranges are wider, particularly in the tropical regions, since, as commented above, the processes connecting emissions to burdens vary greatly between the models. The inter-model spread becomes even clearer when expanding the vertical dimension. This is illustrated by Fig. 4, which shows globally averaged vertical profiles of aerosol MMR change per vertical layer for all species, experiments and models. Differences in the vertical profiles, reflecting differences in vertical transport, between the models can be seen. SPRINTARS and the two CHASER models report among the highest MMR changes. For BC, SPRINTARS have particularly large MMR changes for the *RBUreduced* and *MDEreduced* experiments.

The *SASreduced* experiment (third row, Fig. 4) is associated with the most pronounced upper-level MMR changes, conceivably because this is the region associated with the highest convective activity. Indeed, the average upward moist convective mass flux in the SAS region is more than double what it is in for instance the NAM region (Table 2). Possibly linked to the treatment of convection in the models, we find that GOCART, GEOSCHEMADJOINT and GEOS5 show particularly high upper-level BC changes from emission perturbations in the SAS region. One common denominator for these two models is the use of the above mentioned RAS algorithm, which in a study based on an earlier version of the GEOS model was found to overestimate convective mass transport (Allen et al., 1997). However, while GEOS5 also has large high-altitude burden changes for both OA and $SO_4$ for the *SASreduced* experiment, GOCART and GEOSCHEMADJOINT show very weak high-altitude changes compared to the other models in the $SO_4$ case. Conceivably, wet scavenging, to which $SO_4$ is more subject than BC, is stronger in GOCART than in other models over this region.

Regional increases in aerosol concentrations imposed by emission reductions can be observed for
SPRINTARS and CAMchem, and to a smaller extent also for the CHASER models, GEOS5 and C-
IFS (not shown, but visible in the globally averaged *RBUreduced* and *MDEreduced* plots for OA in
Fig. 4). This occurs mainly for OA and $SO_4$. Aerosol emission reductions may in these models be
influencing the level of oxidants, which would have feedbacks on the concentrations of OA and $SO_4$.
A model study by Shindell et al. (2009) demonstrates the importance of aerosol-gas interactions to the
climate impact of mitigations. They point out that the effect on oxidant changes on $SO_4$ concentrations
are stronger in oxidant-limited regions with high $SO_2$ emissions, and that greater parts of the
industrialized Northern Hemisphere is, in fact, oxidant limited (Berglen et al., 2004).
A contributing cause of the unexpected concentration increases could also be nudging, which is a
simple form of data assimilation that adjusts certain variables of free running climate models to
meteorological re-analysis data – in this case, to constrain the climate to year 2010 meteorology. The
nudging is done differently by the individual model groups. For instance, in SPRINTARS there is no
nudging below altitudes of approximately 300 m, which means that the meteorological field will be
slightly different due to perturbed aerosol effects between the two experiments. This could potentially
involve lower precipitation levels, which would influence the degree of wet removal of particularly
OA and $SO_4$. Nudging has been shown to have the potential to induce forcings that could change the
base characteristics of a model; Zhang et al. (2014) demonstrated using the CAM5 model that nudging
towards reanalysis data resulted in a substantial reduction in cold clouds. Clearly, perturbation
experiments like the ones analyzed in this paper, performed by models with fre-running and nudged
(as opposed to offline) meteorology must be interpreted with caution. A closer investigation of the
cause of the unexpected aerosol concentration increases would be an interesting topic of further
investigations.
We have also calculated regional averages of the MMR change profiles for the regions in Fig. 1 (a),
see Fig. 5. The figure shows the rate of MMR change in a receptor region (colored lines) caused by
emission reductions in a source region (rows), for the three aerosol species (columns). These figures
clearly show the effect of long-range aerosol transport on vertical aerosol profiles: notice for instance
the $SO_4$ burden change profile (rightmost column) for the Arctic (grey), which reaches a maximum at
low altitudes for Russian emission changes (fifth row), but high up for South Asian emission changes
(third row). The HTAP1 study of Shindell et al. (2008) found that upper-troposphere emission-
weighted $SO_4$ and BC concentrations in the Arctic were greatest for emission changes in South Asia
(in the spring) and East Asia (during other seasons), while low-level emission-weighted changes in
Arctic pollution were dominated by emission changes in Europe. While Fig. 5 does not show
emission-weighted numbers, we see the same tendency of nearby source regions (such as Russia)
causing lower-level changes in the Arctic. The large potential of Russian BC emission to influence
Artic climate has been pointed out earlier (Sand et al., 2013b; Stohl, 2006).

### 3.2.1 Aerosol lifetime
Referring to Fig. 2, we have in the bottom row estimated the regional, annually averaged atmospheric
lifetime of the different aerosol species emitted from the six regions, through the relation
$$\tau = \Delta BD(Tg)/\Delta Em(Tg\ day^{-1}) \tag{4}$$
where $\Delta Em$ is the change in emissions on daily timescale within the region (and hence also the global
change), and $\Delta BD$ is the resulting change in global aerosol burden. $SO_4$ has an estimated lifetime of 3-
6 days, except for emissions in the MDE region where the model mean lifetime is 11 days, with an
inter-model spread from 8 (GOCART) to 17 (CHASERre1) days, corresponding to the models with
the lowest and highest $SO_4$ MMR changes, respectively. OA has slightly higher lifetimes; 5-9 days,
except for the MDE regions where the lifetime is above 10 days. This is high compared to the
AeroCom model comparison of Tsigaridis et al. (2014), which found a median global OA lifetime of
5.4 days (range 3.8–9.6 days). Note that fewer models performed the *MDEreduced* and *RBUreduced*
experiments (see Table S-5) and so the estimates for these regions are more uncertain. BC lifetimes are
typically around 11 days for emissions in the MDE and SAS regions and 6-8 days in the other regions,
which is also higher than the 5 days shown by Samset et al. (2014) to be an upper limit for
reproducing remote ocean BC observations. The extended lifetime for aerosols emitted within the SAS
region is likely due to more efficient vertical mixing (see Table 2) and low precipitation except during
the monsoon season. This finding is consistent with previous studies and the longer lifetime is seen
particularly during Northern hemisphere winter (Berntsen et al., 2006). High lifetimes in the MDE
region, particularly for OA and $SO_4$ which are more subject to wet removal, are probably linked to dry
atmospheric conditions (see Table 2).

## 3.3    Radiative forcing changes

In Fig. 6 we show annual and regional averages of the AFE profiles used as input to the RF
calculations (Samset and Myhre, 2011), for the regions in Fig. 1 (a). Underlying calculations were
performed on grid-level using separate profiles for each aerosol species. The global, annual mean BC
AFE in Fig. 6 (a) increases strongly with altitude for all regions, rising from about 400 $Wg^{-1}$ close to
the surface to about 3700 $Wg^{-1}$ at TOA. The reason for this increase is mainly scattering and reflection
from underlying clouds, gases and aerosols, the cumulative amount of which increases with altitude.
This enhances the amount of short wave radiation that the BC aerosol may absorb, and therefore its
radiative impact increases with height. Hence, a given change in BC concentration will have a larger
influence on the total TOA forcing if it occurs at high altitudes than if it occurs at lower altitudes. Note
that the magnitude as well as the exact shape of the profile varies between the regions, depending on
geographic location, climatic factors and surface albedo. For instance, the high surface albedo of the
Arctic or the Middle East renders the radiative impact of the dark BC aerosols, and therefore the AFE
magnitude, particularly high. Also, the vertical increase in the Middle East is less steep than in the
other regions, conceivably due to the lower occurrence of clouds in this area (see Table 2).
Figures 6 (b) and 6 (c) show similar curves for OA and $SO_4$ respectively, with a weaker dependency
on altitude compared to BC. For $SO_4$, a strong maximum close to 900hPa can be seen, mainly related
to humidity and hygroscopic growth (Samset and Myhre, 2011) which significantly enhances the
scattering properties of $SO_4$ aerosols (Haywood et al., 1997; Myhre et al., 2004; Bian et al., 2009), but
which is less relevant for OA.  This is well illustrated by looking at the regionally averaged relative
humidity from MERRA data in Fig. 7, which shows that the Middle East, which has a weak relative
humidity (RH) profile (as well as low average cloud cover; Table 2), is the region with the weakest
$SO_4$ AFE profile. Meanwhile, remote ocean regions typically associated with persistent low-level
clouds (e.g. the South Atlantic or the North/South Pacific) are the areas with the most pronounced $SO_4$
AFE profiles (not shown).
Combining these AFE profiles with aerosol burden changes for each grid cell, month and vertical level
(see Eq. (1)), we obtain direct radiative forcing. Table 3 shows the global mean direct RF, per Tg
emission change, for the three species and six experiments. The forcing ranges between 51.9 and
210.8 $mWm^{-2} Tg^{-1}$ for BC, between -2.4 and -17.9 $mWm^{-2} Tg^{-1}$ for OA, and between -3.6 and -10.3
Wm$^{-2}$ Tg$^{-1}$ for SO$_4$. The HTAP1 study by Yu et al. (2013), which is based on data from nine CTMs and
uses emissions for year 2001 as a baseline, obtained for instance an RF of 27.3 mWm$^{-2}$ Tg$^{-1}$ for BC
from emission reductions in the NAM region. This is substantially lower than our 51.9 mWm$^{-2}$ Tg$^{-1}$ for
the same case, which is related to the host model used to calculate the AFE: As mentioned in Sect.
2.2., we calculate RF based on the OsloCTM2 model, which ranks among the models with highest
AFE for BC in an AeroCom intercomparison study (Myhre et al., 2013). Conversely, GOCART,
which was used to calculate the RF in Yu et al. (2013), had the lowest AFE for BC among the
investigated AeroCom models. The same AeroCom study found that AFE for SO$_4$ was much more
similar between these two host models, and while we find for NAM an SO$_4$ RF of -4.5 mWm$^{-2}$ Tg$^{-1}$,
the number from Yu et al. (2013) is a fairly similar -3.9 mWm$^{-2}$ Tg$^{-1}$. See Samset and Myhre (2015)
for a discussion of the AFE in OsloCTM2.
Mitigations in the Middle East give the largest forcing per Tg emission change for all aerosol species.
The particularly large BC forcing (201.8 mWm$^{-2}$Tg$^{-1}$) is probably related to the region's high surface
albedo, as also found in Samset and Myhre (2015). For OA and SO$_4$, which are more subject to wet
scavenging, the dry atmospheric conditions of the region (Table 2) favor long lifetimes, as shown in
Fig. 2 (bottom row). The opposite can be seen in Russia, for which OA and SO$_4$ forcing is the
weakest; here, the lifetime is the shortest among the regions for these species, and the AFE values are
the smallest (solid blue lines, Fig. 6). Note that while the annually averaged precipitation amount for
2010 was not particularly high in RBU, the region has a high average cloud cover (Table 2 and Fig. 7),
which contributes to wet scavenging. The SAS region also has high RF for all three aerosol species.
For BC, this may be related to the region's high convective activity, which promotes long-range
aerosol transport and therefore high-altitude MMR changes, which due to the BC AFE profile
increases the resulting forcing. A particularly intensive monsoon associated with the strong La Niña
event in 2010 may have contributed to higher convective lifting (and associated effects on the RF) in
this analyses compared to e.g. Yu et al. (2013) or Shindell et al. (2008).
In parentheses in Table 3, we show the relative standard deviation (RSD) values for the RF
calculations – i.e. the sample standard deviation divided by the mean – as a representation of inter-
model spread. In Yu et al. (2013) inter-model differences were also found to be substantial, and one
might expect the spread to be larger due to the large variation in emissions used by the HTAP1
models. However, comparing RSD of emission-weighted RF from Yu et al. (2013) HTAP1 data
(based on their Table 6) to the present HTAP2 data (Table 3), there is no clear tendency that the inter-
model spread for HTAP2 is smaller. In fact, while the RSD for emission-weighted RF for BC
averaged over the four common source regions (NAM, EUR, SAS and EAS) was higher for HTAP1
(0.60) than for HTAP2 (0.37), the opposite was true for the SO$_4$ forcing (RSD of 0.23 and 0.43 for
HTAP1 and HTAP2, respectively). The mixture of models (only CTMs in HTAP1 and both CTMs
and GCMs in HTAP2), the different meteorological years used (2001 in HTAP1 and 2010 in HTAP2),
as well as the fact that HTAP1 region definitions comprised larger areas with much ocean, are
contributing causes that direct comparison of inter-model spread between the two analyses is difficult.
In either case, however, the large ranges in AFE values demonstrates that differences between aerosol
optical properties, treatment of transport and wet removal, and model native meteorology are still
large.  Our results, which are based on simulations using the same set of emissions, also shows notable
inter-model differences. This underlines the importance of model variations in the various aerosol-
related parametrizations – in agreement with previous studies (Kasoar et al., 2016; Textor et al., 2007;
Wilcox et al., 2015).
A more detailed perspective of the global forcing averages of Table 3 can be found in Fig. 8, which
shows the RF, at top-of-atmosphere, estimated to be exerted due to the aerosol abundance change in
each OsloCTM2 model layer. The diversity between models seen in the MMR change in Fig. 4 is
naturally still present, but, in particular for BC, the relative importance of low and high altitudes has
shifted. The strongly increasing BC AFE with altitude dampens BC variability close to the surface,
and emphasizes differences at high altitude. For $SO_4$, the peak in AFE close to 900hPa coincides with
regions of high concentration, leading to increased effective variability in RF exerted close to the
surface. For the same reasons, the particularly large upper-level MMR differences between the models
for the *SASreduced* experiment (Fig. 4) show enhanced RF for BC but dampened for $SO_4$.

## 3.4    Local versus remote impacts of emission mitigation

We move on to quantify how emission mitigations in the six source regions influence radiative forcing
both locally within the source region and in other receptor regions. The leftmost column of Fig. 9
shows the effect of domestic emission reductions on local RF from $SO_4$, OA and BC (Fig. 9 (a), 9 (c)
and 9 (e), respectively). To account for the effect of the large variation in baseline emissions between
the source regions, we have divided the RF by the annually averaged multi-model median emission
change of the source region in question (this gives the forcing efficiency for a given emission change,
but to avoid confusion with the aerosol forcing efficiency, or AFE, profiles used to calculate the RF
we will refer to this quantity as the emission-weighted forcing). Hence, while e.g. EAS has much
larger $SO_2$ emissions than the other regions (Fig. 2) and therefore much larger absolute local forcing
(not shown), the regional difference in the emission-weighted forcing in Fig. 9a is caused by other
factors than the difference in emission levels. For all species, however, the emission-weighted
domestic forcings for the SAS and MDE regions stand out as substantially higher than the other
regions. The numerical values corresponding to Fig. 9 are presented in Tables S-6 through S-8.
Notice that Fig. 9 (a), 9 (c) and 9 (e) have two bars per source region – one solid and one dashed. The
solid bar shows the emission-weighted forcing calculated by Eq. (1), fully accounting for the vertical
aerosol and AFE profile. The hatched bar, however, shows a version calculated by Eq. (2), where we
instead use vertically averaged AFE numbers and total column burden changes (equivalent to the
method that was used for HTAP1 results in Yu et al. (2013)). We can thus study how accounting for
the vertical profiles influences the magnitude of the emission-weighted forcing. For $SO_4$, the vertically
resolved RF calculation gives stronger emission-weighted forcings than the ones using column
burdens: averaged across the regions, treating vertical profiles strengthens $SO_4$ emission-weighted RF
by 25 %. The reason for this is that domestic emission reductions cause changes in atmospheric
aerosol concentrations primarily at low levels, where AFE for $SO_4$ is high. For BC, on the other hand,
RF is reduced by 37 % when accounting for the vertical dimension, because AFE for BC is weak in
the lower atmosphere. For OA, including the vertical information induces only a small increase in
emission-weighted RF of about 8 %. This is unsurprising, given the weak altitude dependence of OA
AFE as shown in Fig. 6.
The rightmost column of Fig. 9 – Fig. 9 (b), 9 (d) and 9 (f) – shows how emission reductions in
different source regions (see x axis) influence the emission-weighted forcing in other receptor regions
(indicated by the colors of the bars clustered above each source region). In general, the extra-regional
forcing is largest for nearby upwind source regions. For instance, for all aerosol species perturbations
in North America have a large effect on the emission-weighted forcing in Europe. Russia, closely
followed by Europe, is the region with the largest influence on the Arctic, and Russia and Europe also
have a strong influence on each other. We similarly find that South Asia has a very large impact on the
emission-weighted forcing in East Asia. However, as noted by Chakraborty et al. (2015) who studied
ozone transport between South and East Asia based on HTAP1 simulations, the influence on South
Asia on East Asia is limited by the onset of the monsoon season, during which the prevailing wind
pattern turns the influence the other way around. In fact, Chakraborty et al. (2015) found that when
focusing on the populated parts of these regions, the emission changes over East Asia had a larger
impact on populated parts of South Asia than vice versa, due to the specific monthly variations of the
meteorological conditions. Another HTAP1 study investigating reductions in methane and ozone
precursor emissions found that among the four source regions NAM, EUR, SAS and EAS, the SAS
region posed the largest emission-weighted influence in terms of radiative forcing, as this region was
located closest to the equator and therefore had the strongest photochemistry, but also due to the
strong vertical mixing during the monsoon season (Fry et al., 2012).
While it is useful to compare extra-regional effects per Tg emission reduction, the potential for sizable
emission reductions is likely to be lower in the regions with the lowest baseline emissions (Table 2).
When we estimated the impact of intercontinental transport by calculating the RERER coefficient (Eq.
3), we therefore use absolute (as opposed to emission-weighted) numbers. Table 4 shows RERER
values for all species and regions. For burden change (top half of Table 4), $SO_4$ RERER is found to be
between 0.32 and 0.76 for the various regions, with high values indicating that a region is strongly
influenced by long-range transport from other regions. OA burden RERER ranges from 0.09 to 0.90,
while BC burden RERER ranges from 0.18 to 0.87. The RERER values are consistent with Chin et al.
(manuscript in preparation), who investigated RERER for HTAP2 data based on surface
concentrations. Due to the experiment design, the source regions are not fully identical between
HTAP1 and HTAP2, so for easier comparison to HTAP1 studies, a version of Table 4 calculated using
the HTAP1 definitions for receptor regions is included in Table S-9. The main features are the same as
in Table 4, but the values are in general higher, as expected since the receptor regions are larger for
HTAP1 than for HTAP2. This difference is most prominent for Europe.
To investigate the impact of the vertical distribution of aerosols, we also calculate RERER for RF
estimated with the vertically resolved AFE distributions (see bottom half of Table 4.) RERER for $SO_4$
and OA are broadly similar for burden change and RF. BC RERER, however, is markedly higher (by
30 %, averaged over all source regions) for RF. This is due to long range transport predominantly
taking place at high altitudes, where BC AFE is strong. Hence any transported BC will have a higher
impact on the RF in remote regions, relative to the source region where it originates close to the
ground. For OA and BC, the RERER for the SAS region is the lowest among the regions, which
means that the region to a lesser extent is influenced by other regions. The RBU and MDE regions
stand out with very high RERER values, indicating that the regions are very sensitive to extra-regional
emission changes. For BC, a high sensitivity of the NAM region to extra-regional emissions is
witnessed by a high RERER value. This sensitivity of North America to emission changes in other
regions has also been noted in other studies, e.g. in a satellite study by Yu et al. (2012).
To visualize the impact of intercontinental transport on the RF that a given receptor region experiences
due to emission reductions in different source regions, we present in Figure 10 a stacked bar plot. For
each species and averaged over the different receptor regions (see x axis), the colors show how much a
20% emission reduction in each of the source region contributes to the summed forcing from all
source regions, in percent. The summed forcing that a receptor region experiences from the six
experiments is given above each bar. Note that as the individual source regions' contribution is
calculated relative to the summed contribution of the six source regions and not relative to a global
emission reduction, as in the calculation of RERER, the numbers in Tab. 4 will be qualitatively but not
quantitatively comparable to this figure. Figure 10 illustrates for instance that the main contributor to
the high RERER value in the NAM region is EAS: for BC, more than 40 % of the summed forcing
originates from emission changes in EAS. The HTAP1 study by Yu et al. (2013) also conluded that
East Asia posed the largest influence on North America for BC RF. However, they also found that for
SO$_4$ RF, South Asia was strongly influenced by emission changes in Europe. This we do not see in our
results, probably because the baseline emissions in Yu et al. (2013) were for year 2001, for which
European SO$_4$ emissions were substantially higher and Indian emissions lower. Other HTAP1 studies
also point to a strong influence of European emission changes: Anenberg et al. (2014) studied impacts
of intercontinental transport of fine particulate matter on human mortality, and found that 17 and 13 %
of premature deaths caused by inhalation of fine particulate matter could be avoided by reducing North
American and European emissions, as opposed to 4 and 2 % for South and East Asia. The main reason
for this, however, was higher downwind populations for the two first regions as opposed to the two
last. Figure 10 shows that domestic mitigations dominate the contribution to the total RF in South and
East Asia, and these are also the regions with the largest forcing contributions to other regions.
However, it is important to note that this relationship is strongly driven by the fact that the baseline
emissions (and hence the 20% emission changes) in EAS and SAS are the largest of all regions, and as
we saw from Fig. 9, the relationship changes when looking at emission-weighted numbers: While Fig.
10 shows e.g. a strong contribution from EAS to the forcing in RBU, Fig. 9 demonstrated that per Tg
emission reduction EUR has a much stronger influence on RBU than EAS.

## 4.  Summary and Conclusions

We have compared RF for the direct aerosol effect from regional 20 % reductions in anthropogenic
aerosol emissions, for ten global climate and chemical transport models participating in the HTAP2
multi-model exercise for the year 2010. We focused on the model experiments simulating emission
reductions in North America, Europe, South Asia, East Asia, Russia/Belarussia/Ukraine and the
Middle East. We find that the globally averaged TOA radiative forcing exerted per Tg of emission
reduction varies between the source regions from 51.9 to 210.8 mWm$^{-2}$ Tg$^{-1}$ for BC, from -2.4 to -17.9
mWm$^{-2}$ Tg$^{-1}$ for OA, and from -3.6 to -10.3 Wm$^{-2}$ Tg$^{-1}$ for SO$_4$. For all species, the globally averaged
emission-weighted forcing from the Middle East was larger than from emission reductions in the other
regions, primarily due to the long lifetime of aerosols originating from this region. For BC, the
emission-weighted forcing was particularly strong due to the high surface albedo of the Middle East.
The second highest values were caused by emission changes in South Asia, due to the high convective
activity, relatively long aerosol lifetime and the low-latitude location. This region, as well as the East
Asian region, also induced the largest regionally averaged emission-weighted forcing in a number of
investigated receptor regions, especially for BC. Mitigations in Europe have strongest impacts on
Russia, the Arctic and the Middle East. Note that relatively long aerosol lifetimes are simulated in this
study, and the BC lifetime is longer than found in models reproducing the vertical profile during the
HIPPO campaigns in the Pacific Ocean (Samset et al., 2014).  A shorter lifetime of BC reduces the RF
of the direct aerosol effect substantially (Hodnebrog et al., 2014).

Although extra-regional mitigations have important contributions to the RF of a given region, the local
influence of emission reductions is for most regions the dominant one. There are however, exceptions:
BC emissions in East Asia are found to be more important to North America than domestic mitigation,
which is consistent with previous findings pertaining the 2000s. A similar feature was found for
Russia for OA and BC; the RF contribution from mitigations in Europe and East Asia outweighs the
region's own influence – at least when mitigations are defined as 20% of the region's baseline
emissions. For the Middle East, OA and BC forcing is dominated by influence from East Asia.
We have also gone beyond previous HTAP studies and investigate the impact of using vertically
resolved concentrations of atmospheric aerosols combined with vertically resolved AFE distributions
when estimating global mean aerosol radiative forcing and intercontinental transport. We find that this
strengthens $SO_4$ RF for all regions, relative to using vertically averaged distributions. BC RF weakens
when using fully resolved distributions, due to a larger weight being put on BC near sources, close to
the ground, where BC AFE is lower. The same feature, only weaker due to a weaker AFE profile, can
be observed for OA. While atmospheric transport of $SO_4$ and OA is only weakly affected, the
influence of inter-continental transport to BC forcing is strengthened by 30 % when accounting for the
vertical aspect, because long-range transport leads primarily to aerosol changes at high altitudes,
where BC AFE is strong.

**Acknowledgement:** This work was supported by the Research Council of Norway through the grants
AC/BC (240372), NetBC (244141) and SLAC. The CESM project is supported by the National
Science Foundation and the Office of Science (BER) of the U. S. Department of Energy. The National
Center for Atmospheric Research is funded by the National Science Foundation. The SPRINTARS is
supported by the supercomputer system of the National Institute for Environmental Studies, Japan, the
Environment Research and Technology Development Fund (S-12-3) of the Ministry of the
Environment, Japan, and JSPS KAKENHI grants 15H01728 and 15K12190. Johannes Flemming's
contribution has been supported by the Copernicus Atmosphere Service. This study also benefitted
from the Norwegian research council projects #235548 (Role of SLCF in Global Climate Regime) and
#229796 (AeroCom-P3).

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

# Figures

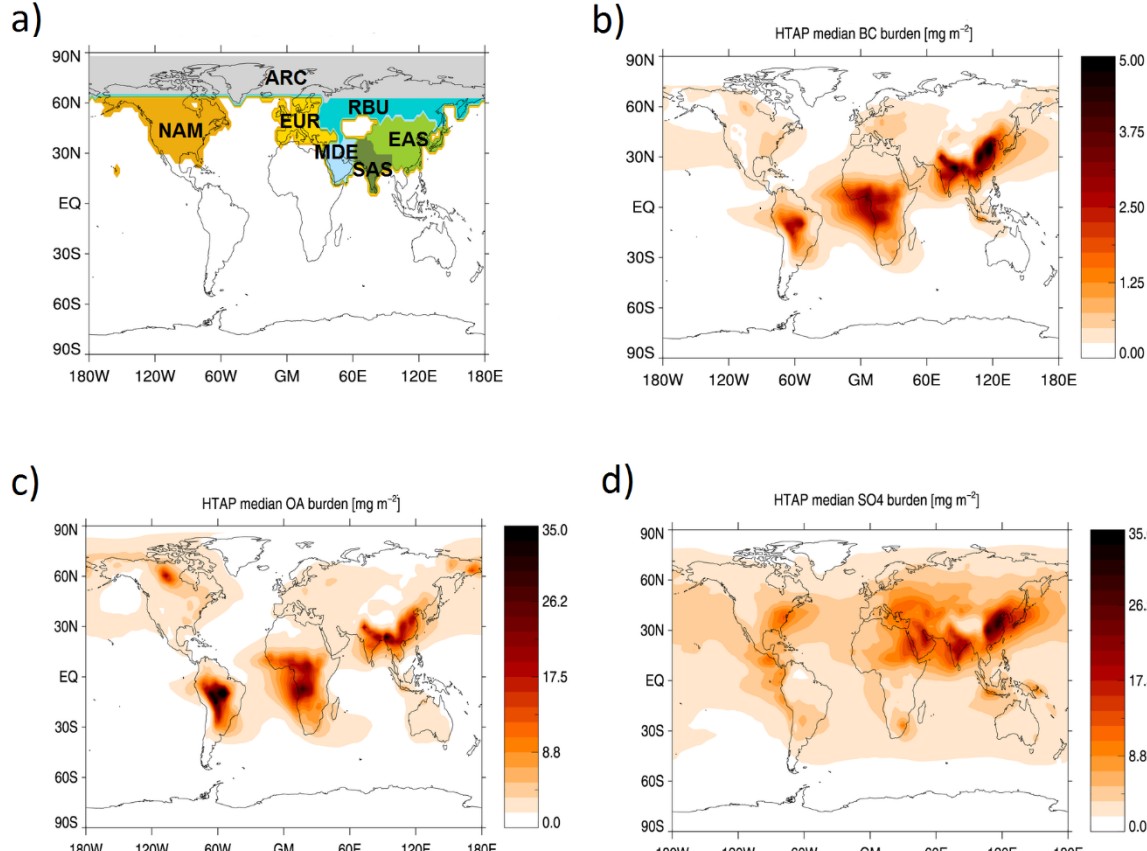

912
913
914

**Figure 1:** a) Regions of focus (NAM: North America; EUR: Europe; EAS: East Asia; SAS: South Asia; RBU; Russia/Belarussia/Ukraine, MDE: Middle East and ARC: Arctic). b), c) and d) show multi-model median (calculated at each grid point), annual mean aerosol load of the *BASE* experiment for BC, OA and SO$_4$, respectively.

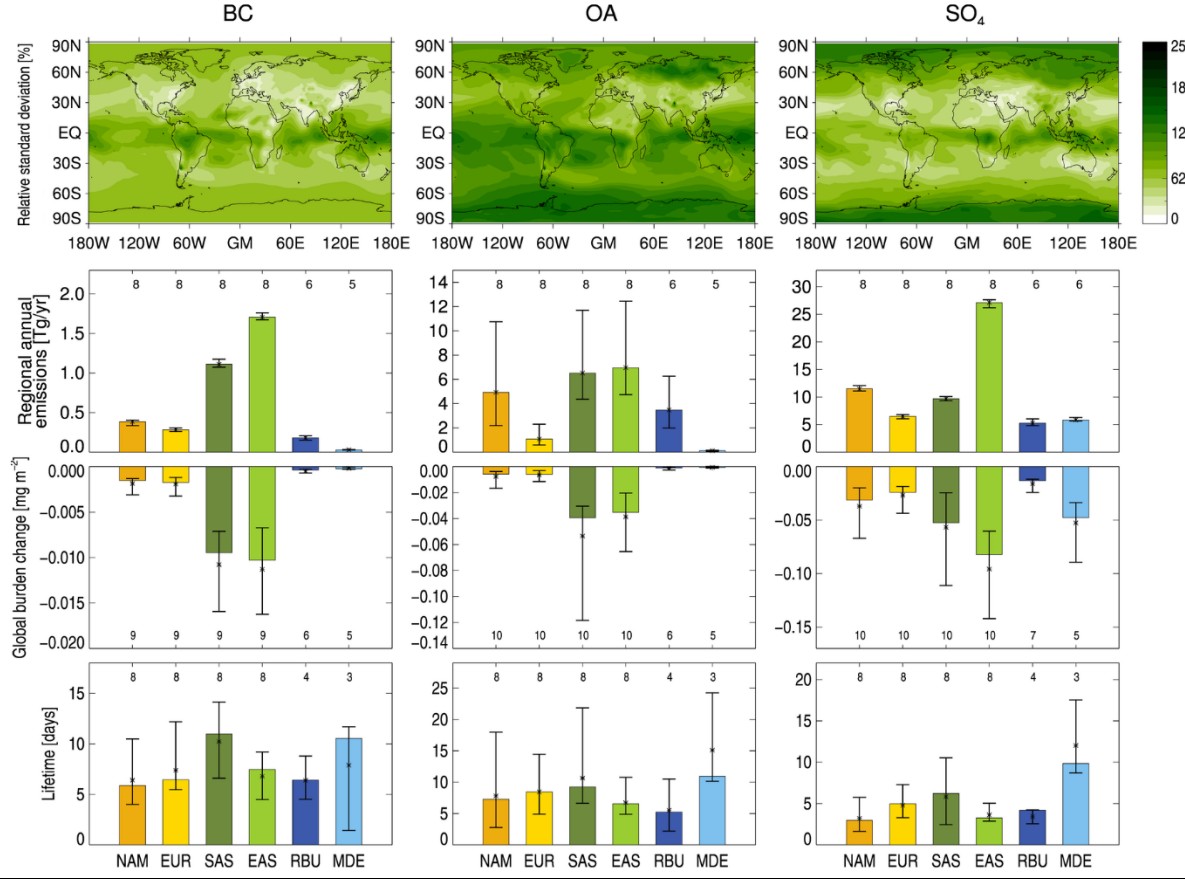

**Figure 2:** Top row: Relative inter-model standard deviation in annual mean aerosol emissions. Second row: Regionally
averaged annual mean aerosol emissions (for $SO_4$, we give $SO_2$ emissions, in Tg $SO_2$), for the source regions shown in Fig. 1.
Numbers are from the *BASE* simulations. Error bars show the maximum and minimum emissions for the sample of models
used here, and the numbers above the bars give the number of models that have data for the given value. Third row: Globally
and annually averaged aerosol burden change for 20 % emission reductions in the indicated region. Numbers are from the
perturbation simulations. Bottom row: Aerosol lifetime, here defined as the global change in burden divided by the global
change in emissions following an emission reduction within a given source region (see main text, Eq. 4).



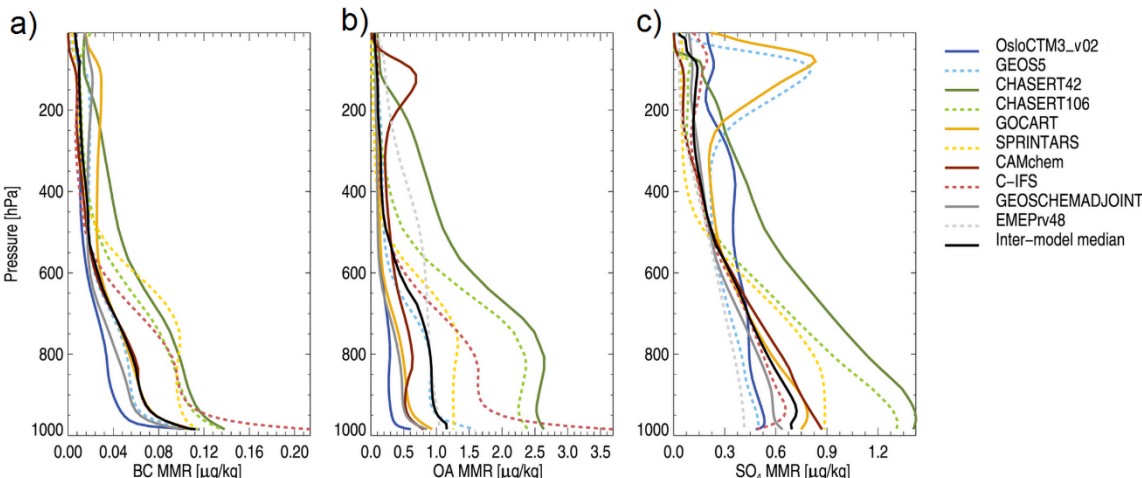

**Figure 3:** Globally and annually averaged mass mixing ratios (MMR) of a) BC, b) OA and c) $SO_4$, for all contributing
models for the *BASE* experiment.

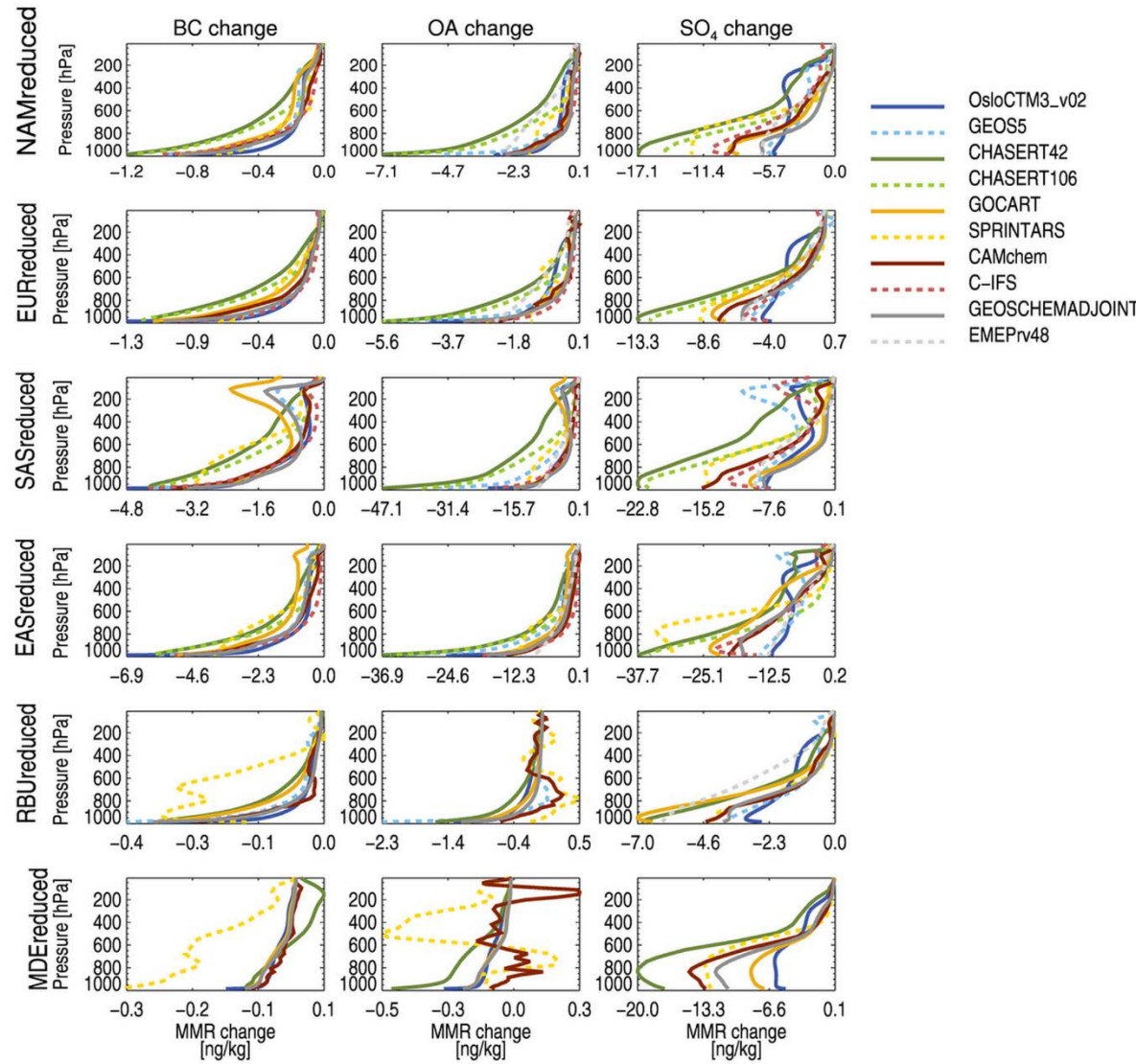

**Figure 4:** Globally averaged change in MMR per model layer, when reducing emissions by 20 % within the region indicated (rows), for all aerosol species (columns). Each line represents one model. See Tables S-2 to S-4 for the total burden changes for all models, experiments and species.

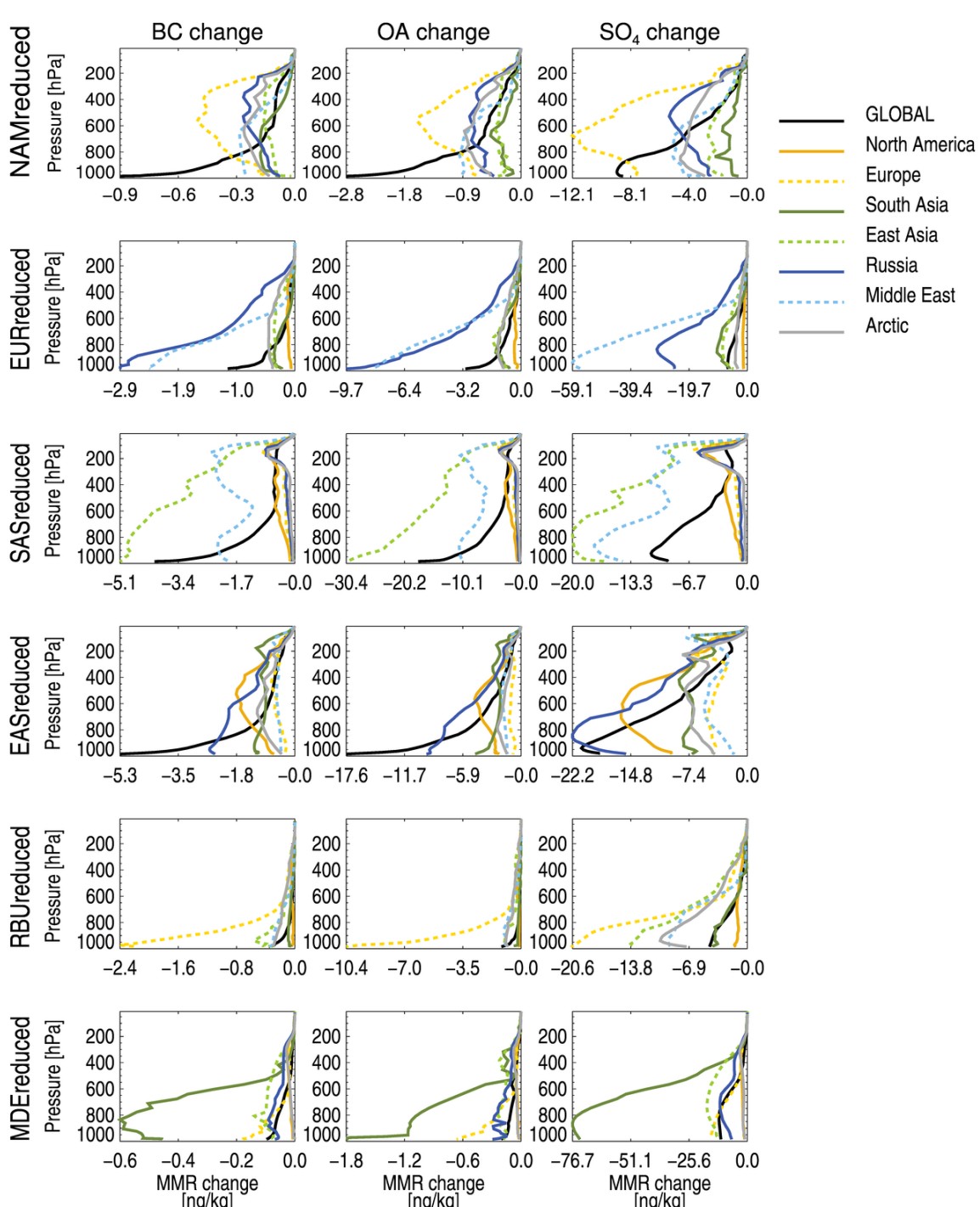

**Figure 5:** Model-averaged aerosol MMR change profiles for different receptor regions (marked by the colors of the lines), for emission reductions in the six source regions (rows) and for BC (first column), OA (middle column) and SO₄ (last column).

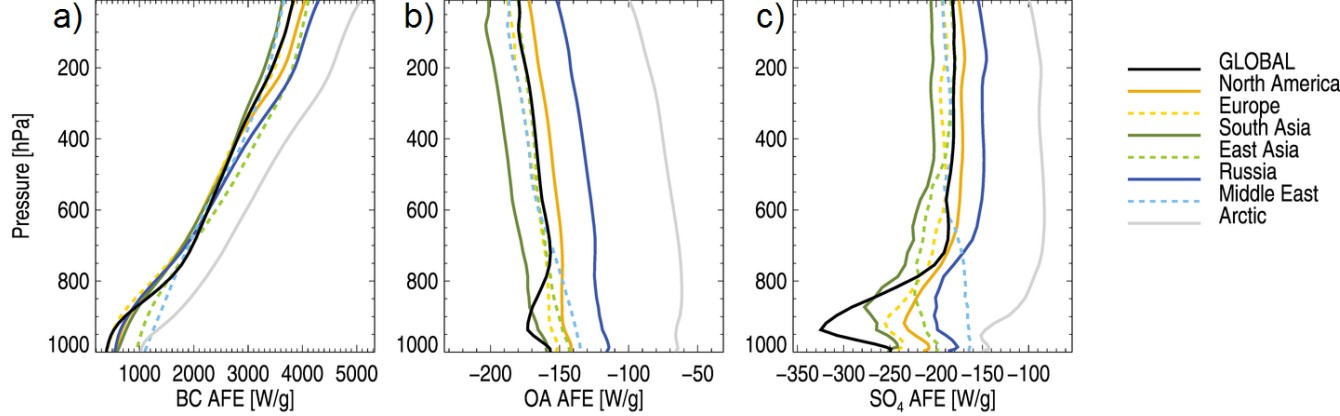



**Figure 6:** Aerosol forcing efficiency profiles, i.e. TOA radiative forcing exerted per gram of aerosol versus altitude,
calculated by the OsloCTM2 model. Black, solid lines indicate global, annual mean profiles. Colored lines show the annual
mean profiles within the regions of Figure 1a.

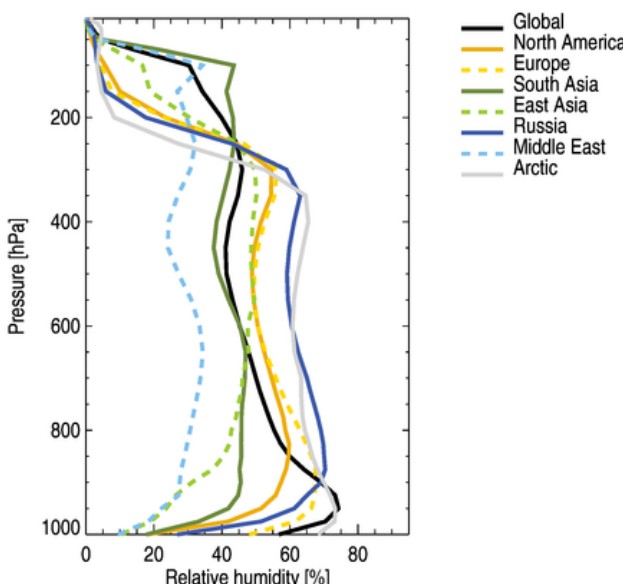


**Figure 7:** Annually averaged relative humidity from MERRA data, for year 2010, for the same regions as in Figure 6.



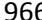

**Figure 8:** Global mean vertically resolved aerosol direct radiative forcing, when reducing emissions by 20 % within the region indicated (rows), for all aerosol species (columns). Each line represents one model. See Tables S-2 to S-4 for individual model results.





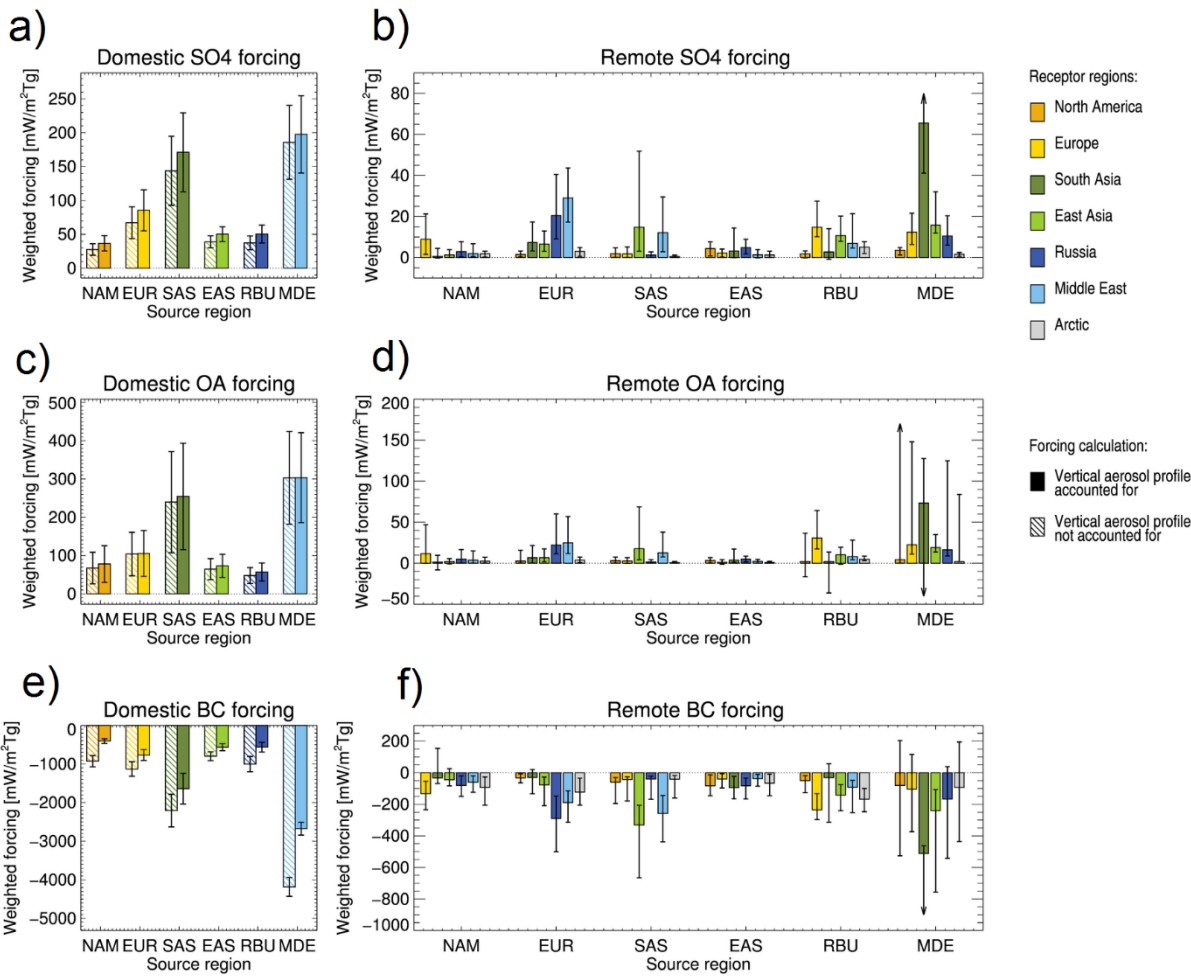


**Figure 9:** Regionally averaged forcing in the six source regions due to domestic emission reductions (leftmost column) and remote forcings averaged over different receptor regions due to emission reductions in the six source regions (rightmost column) for the three aerosol species (tow row: SO₄; middle row: OA; lower row: BC). Forcings are weighed by the emission change in each given source region. The source region in question is marked on the x axis, while the receptor region for which the forcing is averaged is marked by the color of the bar. See Tables S-6 through S-8 for the numbers behind this figure.

980

981

982

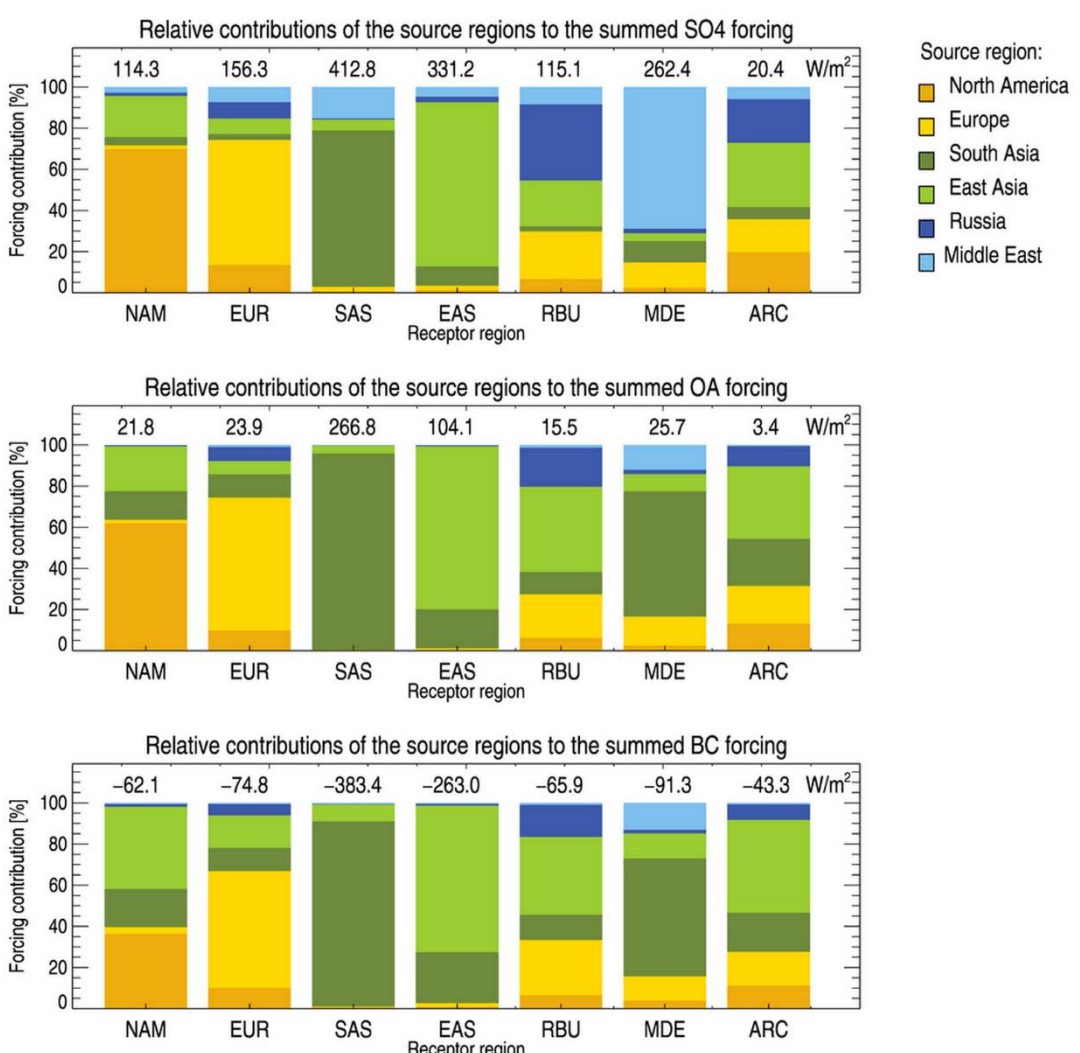

**Figure 10:** Relative contributions of the individual source regions (colors on the bars) to the summed forcing, averaged over each of the receptor regions (given on the x axis and seen in Fig. 1 (a)). The summed forcing that the given receptor region experiences due to emission-reductions in the six source regions is given in numbers above each bar.

# Tables

**Table 1:** Models used for the present study, with relevant information and references.

| | Version | Horizontal resolution | Vert. layers | Meteorology input source | Convection | Reference |
|---|---|---|---|---|---|---|
| **SPRINTARS** | atmosphere: MIROC5.2 | 1.1° x 1.1° | 56 | ECMWF Interim. | The cumulus scheme (Chikira and Sugiyama, 2010) is an entraining plume model, in which the lateral entrainment rate varies vertically depending on the surrounding environment. The cloud base mass flux is determined with a prognostic convective kinetic energy closure. | Watanabe et al. (2010) Takemura et al. (2005) |
| **OsloCTM3_v02** | v02, all aerosol modules from OsloCTM2 | 2.8° x 2.8° | 60 | ECMWF's Integrated Forecast System (IFS) model | The parameterization of deep convection is based on the Tiedke mass flux scheme (Tiedtke, 1989). | Søvde et al. (2012) |
| **GOCART** | v5 2010 | 1.3° x 1.0° | 72 | MERRA | Moist convection is parameterized using archived cloud mass flux fields from MERRA. GCTM convection is parameterized using cloud mass flux information from the relaxed Arakawa-Schubert (RAS) algorithm (Moorthi and Suarez, 1992). | Chin et al. (2000) |
| **C-IFS** | IFS CY40r2 | 0.7° x 0.7° | 54 | Relaxed to ERA-Interim | Tiedtke (1989) shallow convection scheme. | Flemming et al. (2015) |
| **CHASER-T42** | v4.0, MIROC-ESM version | 2.8° x 2.8° | 32 | ERA-Interim (u,v,T) and HadISST | Transport due to advection, convection, and other subgrid-scale mixing are simulated "on-line" by the dynamical component of the CCSR/NIES AGCM. The prognostic Arakawa-Schubert scheme is employed to simulate cumulus convection. | Sudo et al. (2002) |
| **CHASER-T106** | v4.0, MIROC-ESM version | 1.1° x 1.1° | 32 | (as above) | (as above) | Sudo et al. (2002) |
| **CAMchem** | CESM1-CAM4-chemSD | 1.9° x 2.5° | 56 | GEOS5 v5.2 meteorology | Deep convection is parameterized using the Zhang-McFarlane approach (Zhang and McFarlane, 1995), with some modifications, while shallow convection follows Hack et al. (2006) | Tilmes (2016) |
| **GEOS5** | v5 | 1.3° x 1.0° | 72 | MERRA | Convection is based on a modified version of the scheme described by Moorthi and Suarez (1992), which is a relaxed Arakawa-Schubert algoritm (RAS). | Rienecker et al. (2008) Colarco et al. (2010) |
| **GEOSCHEMADJOINT** | v35f | 2.0° x 2.5° | 47 | GEOS-5 (MERRA) | Convective transport in GEOS Chem is computed from the convective mass fluxes in the meteorological archive, as described by Wu et al. (2007), which is taken from GEOS-5 (see above). | Henze et al. (2007) |
| **EMEPrv48** | rv4.8 | 0.5° x 0.5° | 20 | ECMWF's Integrated Forecast System (IFS) model | (see OsloCTM3_v02 above) | Simpson et al. (2012) |

**Table 2:** Regionally averaged burdens and climatological features for the six source regions. Burdens are multi-model median, annually averaged values for the *BASE* experiment with one multi-model standard deviation in parenthesis. Convective mass flux (for the layers between 1000 and 500 hPa), precipitation and cloud cover represent regionally and annually averaged values for 2010 from the Modern-Era Retrospective analysis for Research and Applications (MERRA) reanalysis data set.

| | Region name | BC burden [mgm$^{-2}$] | OA burden [mgm$^{-2}$] | SO$_4$ burden [mgm$^{-2}$] | Convective mass flux [kgm$^{-2}$] | Precipitation [mm/day] | Cloud cover [%] |
|---|---|---|---|---|---|---|---|
| **NAM** | North America | 0.36 (± 0.09) | 3.86 (± 3.45) | 3.55 (± 1.28) | 3980 | 1.92 | 55 |
| **EUR** | Europe | 0.39 (± 0.09) | 2.70 (± 1.83) | 5.44 (± 1.43) | 4774 | 1.89 | 53 |
| **SAS** | South Asia | 1.85 (± 0.36) | 14.57 (± 7.67) | 11.34 (± 3.57) | 9769 | 3.34 | 43 |
| **EAS** | East Asia | 1.25 (± 0.26) | 7.48 (± 4.17) | 9.16 (± 2.43) | 4105 | 1.89 | 46 |
| **RBU** | Russia | 0.29 (± 0.09) | 2.84 (± 2.71) | 4.58 (± 2.05) | 2741 | 1.44 | 63 |
| **MDE** | Middle East | 0.41 (± 0.12) | 3.43 (± 3.53) | 11.54 (± 3.48) | 1247 | 0.41 | 23 |

**Table 3:** Globally averaged radiative forcing from the six main experiments, weighed by the emission change for the given source region. Relative one standard deviations (representing multi-model variation) are given in parentheses.

| | BC [mWm$^{-2}$Tg$^{-1}$] | OA [mWm$^{-2}$Tg$^{-1}$] | SO$_4$ [mWm$^{-2}$Tg$^{-1}$] |
|---|---|---|---|
| **NAMreduced** | 51.9 (± 0.4) | -7.9 (± 0.8) | -4.5 (± 0.5) |
| **EURreduced** | 55.2 (± 0.4) | -6.8 (± 0.6) | -5.6 (± 0.4) |
| **SASreduced** | 93.8 (± 0.4) | -10.2 (± 0.6) | -7.9 (± 0.5) |
| **EASreduced** | 54.5 (± 0.3) | -5.1 (± 0.5) | -4.4 (± 0.3) |
| **RBUreduced** | 78.3 (± 0.6) | -2.4 (± 2.2) | -3.6 (± 0.3) |
| **MDEreduced** | 201.8 (± 1.6) | -17.9 (± 0.4) | -10.3 (± 0.7) |

**Table 4:** Response to Extra-Regional Emission Reductions (RERER), averaged over the 10 participating models, ± one standard deviation representing inter-model spread. A high RERER value means that the given region is very sensitive to extra-regional emission reductions. The top table shows RERER for column aerosol burdens, the bottom table shows RERER for direct radiative forcing (DRF) calculated using vertically, spatially and temporally resolved AFE profiles.

| Burden change | NAM | EUR | SAS | EAS | RBU | MDE |
|---|---|---|---|---|---|---|
| **BC** | $0.51 \pm 0.13$ | $0.37 \pm 0.06$ | $0.12 \pm 0.03$ | $0.21 \pm 0.05$ | $0.83 \pm 0.04$ | $0.87 \pm 0.04$ |
| **OA** | $0.49 \pm 0.19$ | $0.41 \pm 0.08$ | $0.09 \pm 0.03$ | $0.24 \pm 0.06$ | $0.82 \pm 0.06$ | $0.90 \pm 0.06$ |
| **SO$_4$** | $0.46 \pm 0.14$ | $0.54 \pm 0.09$ | $0.36 \pm 0.04$ | $0.32 \pm 0.07$ | $0.75 \pm 0.06$ | $0.46 \pm 0.08$ |
| **DRF** | NAM | EUR | SAS | EAS | RBU | MDE |
| **BC** | $0.69 \pm 0.11$ | $0.57 \pm 0.10$ | $0.18 \pm 0.04$ | $0.37 \pm 0.06$ | $0.89 \pm 0.03$ | $0.91 \pm 0.03$ |
| **OA** | $0.46 \pm 0.18$ | $0.46 \pm 0.08$ | $0.09 \pm 0.02$ | $0.27 \pm 0.06$ | $0.83 \pm 0.07$ | $0.91 \pm 0.06$ |
| **SO$_4$** | $0.41 \pm 0.12$ | $0.53 \pm 0.08$ | $0.34 \pm 0.04$ | $0.31 \pm 0.07$ | $0.73 \pm 0.05$ | $0.47 \pm 0.08$ |