# Peer review of "Table S-1: Model treatment of OA/OC and aerosol schemes."

_Atmospheric Chemistry and Physics, 2016_

## Referee Comment (RC1) · Anonymous Referee #1 · 30 Jun 2016

This is a well-written study that evaluates changes in radiative forcing caused by 20% reductions in regional emissions of aerosols and aerosol precursors (BC, OC, and SO2), using 10 models contributing to the HTAP2 project. The paper includes helpful discussion on spatial variability in forcing efficiency and forcing perturbations, as well as comparisons of contributions to forcing from local and remote emissions and an evaluation of the importance of using vertically-resolved aerosol fields for achieving accuracy in total aerosol forcing. The scope of the paper is somewhat narrow, as it only describes direct aerosol forcing from the models, but the evaluation of radiative forcing from 10 models contributing to a community experiment certainly makes this study worthy of publication. I have minor suggestions for revisions, as described below.

[Figure]

General comments:

Was direct aerosol forcing calculated internally by any of the contributing aerosol models? If so, it would be very helpful to compare these forcing estimates with those obtained by running the aerosol fields through the authors' radiative kernels. This could offer an indication of how much additional variability in forcing might be expected from different model cloud fields, assumed aerosol optical properties, and radiative transfer codes.

Minor comments:

Abstract: It would be helpful to include some of the overarching quantitative results in the abstract, namely the model-mean changes (and perhaps inter-model standard deviation) in global radiative forcing resulting from the emissions perturbations. I view these numbers as headline results from the study that should be reported in the abstract.

Introduction (and last sentence of abstract): I suggest mentioning that although BC radiative efficiency increases with BC altitude, the associated surface temperature change does not, and can even become opposite of the sign of TOA forcing when the BC is located at sufficiently high altitude.

line 75: "on" -> "of"

line 105-109: How do the HTAP2 results compare with these HTAP1 results?

Section 2.1, line 132: Please list the global annual emissions of each species, as represented in the inventories applied.

Section 2.1, paragraph 2: Please mention here that all models use prescribed meteorology (rather, e.g., than prescribed SSTs with online meteorology), assuming this is indeed true.

lines 155-158: How much error does this interpolation technique introduce to the estimates of column-integrated aerosol abundance (burden), as opposed to using each host model's pressure fields self-consistently with their aerosol fields? It is probably small, but worth mentioning. (And why were these calculations done with OsloCTM pressure/mass fields instead of the native model's fields?)

Section 2.2, paragraph 1: What spectral resolution (or how many spectral bands) was applied in the radiative transfer calculations?

line 168: Did all models provide separate mixing ratios for "aged" and "non-aged" BC? If not, how did you partition the BC fields into these two components for the radiative forcing calculations?

line 180-181: I assume that the forcings presented here are instantaneous forcings, rather than adjusted or effective forcings, but please clarify this.

line 194: Are the vertically-averaged AFEs weighted by aerosol mass? (presumably so).

Section 3 / Figure 1: I think it would also be quite useful to show/describe the inter-model variability (e.g., standard deviation or normalized standard deviation) in aerosol burden. This would provide a nice depiction to readers of where the models tend to differ from each other the most. Deviation plots could either be included in Figure 2 or added as a separate figure.

Lines 301-312: Could non-linear chemical processes provide an alternative explanation for this odd behavior of increasing aerosol concentrations in response to emissions reductions?

lines 394-402: Are the intermodel differences in radiative forcing larger or smaller than the differences found by Yu et al (2013) for HTAP1 simulations? Presumably they are smaller because of the use of identical emissions data in HTAP2, but it would be helpful to provide a semi-quantitative comparison of the inter-model variability between these two studies.

[Figure]

line 499: "...13% of global deaths" - Is this 13% of the deaths caused by inhalation of fine particulate matter? Please clarify.

Figure 1: Do the median fields shown here represent fields from a single (median) model, or is the median computed at each gridcell from all models? Please clarify.

Figure 2: Are the SO2 emissions reported in Tg of S or Tg of SO2? Please clarify.

Figure 8: It seems the units here should have a vertical component, e.g., mW/m2/Pa or mW/m2/m or mW/m2/layer. Is this so? Otherwise, how does one obtain the typical column radiative forcing (W/m2) from these vertical profiles? Please clarify.

Table 2: It would be helpful to also include the multi-model standard deviations for convective mass flux, precipitation, and cloud, if at all possible.
* * *

---

## Referee Comment (RC2) · Anonymous Referee #2 · 31 Jul 2016

The manuscript by Stjern et al. examines the local and remote influences of aerosol and aerosol precursor emissions on regional atmospheric aerosol abundances and radiative forcing. This is achieved using results from 20% regional emission reduction experiments performed as part of the Hemispheric Transport of Air Pollution Phase 2 (HTAP) exercise in conjunction with pre-calculated aerosol forcing efficiencies. The authors document the various responses, and towards the end they additionally explore the influence of the vertical distribution of aerosols on the results. The manuscript is well written and certainly within the scope of Atmospheric Chemistry and Physics. Even though there are no particularly novel findings in it, the fact that it documents and thoroughly discusses the results of these important new multi-model experiments

makes it worth publishing, following some (mostly minor) corrections that I suggest below.

SPECIFIC COMMENTS:

Abstract: I suggest that the authors clearly state that in most cases the local influence is the dominant, though with important contributions from remote regions. Then to proceed with outlining specific features, as the already do.

Page 2, Line 69: What is meant by "efficiency"? Presumably "radiative forcing efficiency". Please clarify.

Page 3, Lines 105-108: Presumably the three numbers refer to the three different species. But which region do they refer to?

Page 4, Line 131: Does the climate correspond to the specific year 2010 in the models, or to climatic conditions representative of years "around" 2010. Please specify, and if the former, please discuss how the choice of a single year may affect the conclusions.

Page 4, Line 155: I suggest changing "to abundance" to "to column abundance", as "abundance" is a general term that can refer to pretty much anything (including the MMRs).

Sect. 2.2: It should be clearer to a reader not familiar with the Samset and Myhre (2011) manuscript how the OsloCTM2 was utilized for calculating those AFEs. Initially it is stated that a radiative transfer code is used for calculating the AFEs, with the OsloCTM2 providing the background aerosols. However, later it is mentioned as "the host model", and that "the absolute RF will be influenced by the mean efficiency of the host model". Please clarify.

Page 4, Line 162: Suggest changing to "emission and subsequent concentration reductions".

Page 4, Line 165: What does "a series of simulations" mean here? Please explain.

Page 4, Line 168: Please provide reference justifying this adjustment.

Page 4, Line 169: What type of scaling was used? Reference?

Page 5, Line 175: I would say that it would be useful to give a brief summary of the impact of using a single model's kernel in this manuscript too, given how central AFEs are for the results presented here. No need to be long – a few sentences would suffice.

Page 5, Line 177: I presume by "resulting" it is meant "modelled"? If so, please change, clearly mentioning that this is from the new HTAP2 simulations.

Page 5, Line 178: Please remind the reader that "time" here corresponds to the month of the year.

Page 6, Lines 214-215: Suggest rephrasing the part of the sentence after the second comma with "which gives an estimate of inter-continental transport in two dimensions (ignoring the vertical)".

Page 8, Lines 301-312: This is a peculiar feature and needs some further explanation. It is a bit too hand-wavy to say that nudging may be responsible. If so, why would it mainly appear in SO4 and OA, but not BC? Maybe it has to do with effects of aerosol emission reductions on oxidants in the models? SO4 and OA would be affected by this, but not BC.

Page 8, Lines 315-319: Is this in agreement with what other studies that examined long-range transport of pollution to the Arctic (e.g. Shindell et al., 2008) have found? Worth mentioning.

Figure 6: I suggest reminding the reader in the caption that these are inferred from one model, and which model this is.

Page 9, Line 346: I suggest clarifying that panel (a) is for BC. This is more important than mentioning the panel.

Page 9, Lines 354-356: Worth mentioning that the vertical increase in the Middle East

is not as steep, presumably due to the lower occurrence of clouds in this area.

Page 10, Lines 387-388: Could it also be the lower insolation in this region?

Page 10, Line 402: Worth citing the recent paper by Kasoar et al. (2016) here, as it also discusses thoroughly the causes of diversity in three different models when it comes to simulating climate impacts of identical regional SO2 emission perturbations.

Table 4: I may be missing something, but shouldn't the values in Table 4 be consistent with the values in Figure 10? I cannot see this fully being true: For example, for the first bar in Fig. 10 - representing North America - the domestic influence seems to be responsible for much more than half of the SO4 forcing, but then in Table 4 it appears as if it is just above 60%. Please check consistency (for the whole table) and/or explain.

Table 4: I suggest mentioning what the range indicated next to the means represents.

Page 13, Line 520: Please add "surface" before "albedo".

Page 13, Lines 526-530: As mentioned in the abstract as well, I suggest that you clearly state at the beginning of this paragraph that in most cases the local influence is the dominant, though with important contributions from remote regions. And then proceed with outlining the cases where remote is stronger than local (as is done already).

Page 13, Lines 531-533: In the first sentence of this paragraph, it is mentioned that the effect of "vertically resolved concentrations" is examined, while in the next sentence it is stated that "Using vertically resolved AFE distributions strengthens...". Which of the two is examined, the influence of smoothed concentrations or of smoothed AFEs? Earlier it is mentioned that the effect of both is examined. Please clarify.

REFERENCES:

Kasoar, M., Voulgarakis, A., Lamarque, J.-F., Shindell, D. T., Bellouin, N., Collins, W. J., Faluvegi, G., and Tsigaridis, K. (2016), Regional and global temperature response to anthropogenic SO2 emissions from China in three climate models, Atmos. Chem.

Phys., 16, 1–20, doi:10.5194/acp-2015-1017.

Shindell, D. T., Chin, M., Dentener, F., Doherty, R. M., Faluvegi, G., Fiore, A. M., Hess, P., Koch, D. M., MacKenzie, I. A., Sanderson, M. G., Schultz, M. G., Schulz, M., Stevenson, D. S., Teich, H., Textor, C., Wild, O., Bergmann, D. J., Bey, I., Bian, H., Cuvelier, C., Duncan, B. N., Folberth, G., Horowitz, L. W., Jonson, J., Kaminski, J. W., Marmer, E., Park, R., Pringle, K. J., Schroeder, S., Szopa, S., Takemura, T., Zeng, G., Keating, T. J., and Zuber, A. (2008), A multi-model assessment of pollution transport to the Arctic, Atmos. Chem. Phys., 8, 5353-5372, doi:10.5194/acp-8-5353-2008.
* * *
* * *

---

## Author Comment (AC1) · 16 Sep 2016

Was direct aerosol forcing calculated internally by any of the contributing aerosol models? If so, it would be very helpful to compare these forcing estimates with those obtained by running the aerosol fields through the authors' radiative kernels. This could offer an indication of how much additional variability in forcing might be expected from different model cloud fields, assumed aerosol optical properties, and radiative transfer codes.

Response: This was indeed considered, as we agree that much could be learned from such an analysis. No participating model group initially performed the RF calculations, but we were in contact with one group on adding one sensitivity test. In the end,

however, they were not able to deliver the results and we decided to proceed without the analysis. For a full analysis where both native mode RF and kernel estimates was available, however, see Samset et al. 2013, ACP, "Black carbon vertical profiles strongly affect its radiative forcing uncertainty", Figure 1. There, it was found that for BC, between 20 and 50% of the variability can be attributed to vertical profiles alone, with the rest being due to a combination of optical properties, horizontal transport and differences in cloud fields. Also note that Stier et al. 2013 (ACP, "Host model uncertainties in aerosol radiative forcing estimates: results from the AeroCom Prescribed intercomparison study ") investigated model uncertainty in direct RF for twelve Aero-Com models and found substantial diversity in both clear- and all-sky RF even when aerosol radiative properties were prescribed. For HTAP2, a followup study with 2-3 models may be performed later.

Abstract: It would be helpful to include some of the overarching quantitative results in the abstract, namely the model-mean changes (and perhaps inter-model standard deviation) in global radiative forcing resulting from the emissions perturbations. I view these numbers as headline results from the study that should be reported in the abstract.

Response: We absolutely agree, and we have tried several approaches to this in earlier manuscript versions. However, the "main results" comprise forcings from emission changes in six different regions, for three different species. Obviously, a listing of eighteen numbers in the abstract is less than ideal. We therefore chose to list the ranges of RF resulting from emission reduction in the six regions, as the numbers vary so much between the regions.

Introduction (and last sentence of abstract): I suggest mentioning that although BC radiative efficiency increases with BC altitude, the associated surface temperature change does not, and can even become opposite of the sign of TOA forcing when the BC is located at sufficiently high altitude.

Response: This is an important point, and we now comment on this in both abstract and introduction. In the abstract, we added the text: "In the present study, it has only been possible to estimate the effects of emission reductions on instantaneous top-of-atmosphere RF from the direct effect. As the climate response efficiency to aerosols, in particular that of BC, also depends on altitude, we do not extend our analysis to estimates of temperature or precipitation changes." In the Introduction, we added the text: "Previous studies have shown that the relationship between instantaneous RF, which is what we estimate here, and the surface temperature change following a change in BC also depends on the altitude of the BC. Although not found in all studies (Ming et al., 2010), there is a tendency that BC inserted near the surface causes warming, whereas BC near the tropopause and in the stratosphere causes cooling (Ban-Weiss et al., 2012; Samset and Myhre, 2015; Sand et al., 2013a; Shindell and Faluvegi, 2009). This is mainly related to the semi-direct effect of BC, which causes a negative RF through suppression of cloud formation by enhancing atmospheric stability, and which is not accounted for when calculating the instantaneous forcing via radiative kernels. It is beyond the scope of study to calculate climate change in terms of surface temperature change, and we stress that a positive/negative estimate of direct RF here should not be translated directly into warming or cooling."

line 75: "on" -> "of"

Response: The word is changed.

line 105-109: How do the HTAP2 results compare with these HTAP1 results?

Response: We do provide some comparison (Section 3.3, third paragraph) of emission-weighted radiation changes between HTAP1 (Yu et al.) and HTAP2. HTAP1 and HTAP2 had different sets of contributing models, and we do comment on this and other causes of HTAP1-HTAP2 differences in this and the following sections.

Section 2.1, line 132: Please list the global annual emissions of each species, as represented in the inventories applied.

Response: The emission numbers have been included in this sentence.

Section 2.1, paragraph 2: Please mention here that all models use prescribed meteorology (rather, e.g., than prescribed SSTs with online meteorology), assuming this is indeed true.

Response: That is correct, and we now specify this.

lines 155-158: How much error does this interpolation technique introduce to the estimates of column-integrated aerosol abundance (burden), as opposed to using each host model's pressure fields self-consistently with their aerosol fields? It is probably small, but worth mentioning. (And why were these calculations done with OsloCTM pressure/mass fields instead of the native model's fields?)

Response: The reason that the calculations are performed with fields from one model, is primarily that the input data was not readily available from the other models. Further, as we interpolate to a common vertical dimension for comparability anyway, this method entails the fewest interpolations. For previous analyses with AeroCom Phase II data, the interpolation has shown to change column burdens by less than 1%.

Section 2.2, paragraph 1: What spectral resolution (or how many spectral bands) was applied in the radiative transfer calculations?

Response: Four short wave bands. The text has been updated.

line 168: Did all models provide separate mixing ratios for "aged" and "non-aged" BC? If not, how did you partition the BC fields into these two components for the radiative forcing calculations?

Response: We assume the same mixing ratio as in OsloCTM2 for all models. This information is now added at this location.

line 180-181: I assume that the forcings presented here are instantaneous forcings, rather than adjusted or effective forcings, but please clarify this.

Response: Yes, the forcings are instantaneous, and this is now stressed.

line 194: Are the vertically-averaged AFEs weighted by aerosol mass? (presumably so).

Response: Yes. They stem from separate calculations where a realistic (i.e. stemming from AeroCom Phase II emission) vertical profile has been used. Also, on practice, this would affect the overall scaling, but not the analysis of change in inter-model variability (for comparison to Yu et al. 2011, for instance) so long as the same 2D field is used systematically.

Section 3 / Figure 1: I think it would also be quite useful to show/describe the intermodel variability (e.g., standard deviation or normalized standard deviation) in aerosol burden. This would provide a nice depiction to readers of where the models tend to differ from each other the most. Deviation plots could either be included in Figure 2 or added as a separate figure.

Response: This is indeed relevant information. We have created relative standard deviation plots for the three species and included them in Figure 2 as suggested. In general, the models disagree the most over the tropics and over the poles, and we comment upon this in the text.

Lines 301-312: Could non-linear chemical processes provide an alternative explanation for this odd behavior of increasing aerosol concentrations in response to emissions reductions?

Response: Yes it could. We have previously been in close contact with the modelling groups of the models showing this unexpected increase, and have had no suggestions as to the cause, other than the nudging. This is, however, only a suggested cause, and the alternative explanation suggested by the reviewer is just as valid. We have therefore extended this section, including a few sentences on oxidation feedbacks.

lines 394-402: Are the intermodel differences in radiative forcing larger or smaller than

the differences found by Yu et al (2013) for HTAP1 simulations? Presumably they are smaller because of the use of identical emissions data in HTAP2, but it would be helpful to provide a semi-quantitative comparison of the inter-model variability between these two studies.

Response: We agree that such a comparison would be helpful. Yu et al. (2013) do provide emission-weighted standard deviations of emission-weighed forcing, which we have translated to relative standard deviations and compared to our numbers from Table 3. The different meteorological years used for the two analyses, as well as the different set of models (only CTMs In Yu et al and a mix of CTMs and GCMs here) and the different region definitions, precludes a proper quantitative comparison, but we do give the numbers as well as a short discussion at this place in the text.

line 499: "...13% of global deaths" - Is this 13% of the deaths caused by inhalation of fine particulate matter? Please clarify.

Response: Yes it is – this is now clarified in the text.

Figure 1: Do the median fields shown here represent fields from a single (median) model, or is the median computed at each gridcell from all models? Please clarify.

Response: Median fields are calculated in each grid point – this is now clarified in the figure caption.

Figure 2: Are the SO2 emissions reported in Tg of S or Tg of SO2? Please clarify.

Response: SO2 emissions are reported in Tg SO2, as now stated in the figure caption.

Figure 8: It seems the units here should have a vertical component, e.g., mW/m2/Pa or mW/m2/m or mW/m2/layer. Is this so? Otherwise, how does one obtain the typical column radiative forcing (W/m2) from these vertical profiles? Please clarify.

Response: Yes, the unit is supposed to read mW/m2/layer – this is now fixed in the figure.

[Figure]

Table 2: It would be helpful to also include the multi-model standard deviations for convective mass flux, precipitation, and cloud, if at all possible.

Response: That would indeed have been interesting to see, but this information is regretfully not available to us.

---

## Author Comment (AC2) · 16 Sep 2016

Abstract: I suggest that the authors clearly state that in most cases the local influence is the dominant, though with important contributions from remote regions. Then to proceed with outlining specific features, as the already do.

Response: We agree that this should be stressed, and have now added this to the abstract.

Page 2, Line 69: What is meant by "efficiency"? Presumably "radiative forcing efficiency". Please clarify.

Response: That is correct. This is now clarified.

[Figure]

Page 3, Lines 105-108: Presumably the three numbers refer to the three different species. But which region do they refer to?

Response: This was indeed unclear. The numbers are, as given in Yu et al, averages over all four regional emission-reduction experiments, and we have tried to express this more clearly now.

Page 4, Line 131: Does the climate correspond to the specific year 2010 in the models, or to climatic conditions representative of years "around" 2010. Please specify, and if the former, please discuss how the choice of a single year may affect the conclusions.

Response: The climate corresponds to the specific year 2010, and the reviewer is of course right that the results of this paper will be affected by that. We have added a couple of sentences discussing this.

Page 4, Line 155: I suggest changing "to abundance" to "to column abundance", as "abundance" is a general term that can refer to pretty much anything (including the MMRs).

Response: We agree that the specification is necessary, and this is now fixed.

Sect. 2.2: It should be clearer to a reader not familiar with the Samset and Myhre (2011) manuscript how the OsloCTM2 was utilized for calculating those AFEs. Initially it is stated that a radiative transfer code is used for calculating the AFEs, with the OsloCTM2 providing the background aerosols. However, later it is mentioned as "the host model", and that "the absolute RF will be influenced by the mean efficiency of the host model". Please clarify.

Response: This section has been revised and clarified. It now reads: "In order to estimate the radiative forcing resulting from the emission and subsequent concentration reductions simulated by the HTAP2 experiments, we utilize precalculated 4D distributions of aerosol forcing efficiency (AFE), which is defined as the RF per gram of a given aerosol species. For the three aerosol species, AFE was calculated for each grid

cell and month by inserting a known amount of aerosol within a known background of aerosols and clouds, for each model layer individually, and calculating the resulting radiative effect using an 8-stream radiative transfer model with four short wave spectral bands (Stamnes et al., 1988). I.e. the model was used to calculate the response to a change in aerosol concentration at a given altitude, and run for a whole year to capture seasonal variability. The simulations for different model layers were then combined into a set of radiative kernels, one for each aerosol species. For the radiative transfer calculations aerosol optical properties were derived from Mie theory. The absorption of aged BC was enhanced by 50% to take into account external mixing, as suggested by Bond and Bergstrom (2006), and for all models we assume the same mixing ratio between aged and non-aged BC as in OsloCTM2. Hygroscopic growth of SO4 was included, scaling with relative humidity according to Fitzgerald (1975). See Myhre et al. (2004) for a discussion on the impacts of this choice. For OA, purely scattering aerosols are assumed. Background aerosols were taken from simulations using OsloCTM2. See Samset and Myhre (2011) for details, but note that all numbers have been updated since that work, taking into account recent model improvements (Samset and Myhre, 2015). The resulting AFE profiles, averaged over the individual regions from Fig. 1 (a), is presented in Sect. 3.3. For a full discussion on the impact on radiative forcing from using a single model kernel, see Samset et al. (2013). Briefly, multi-model average forcing becomes representative of that of the most model, including cloud fields and optical properties, while the variability around this value is indicative of the impact of differences in 3D aerosol burdens. The resulting reduction in multi-model relative standard deviation depends on the regional and vertical differences in AFE, but is generally less than 20%."

Page 4, Line 162: Suggest changing to "emission and subsequent concentration reductions".

Response: This is now changed.

Page 4, Line 165: What does "a series of simulations" mean here? Please explain.

Response: See the revised section above.

Page 4, Line 168: Please provide reference justifying this adjustment.

Response: A reference to Bond and Bergstrom 2006 ("Light Absorption by Carbonaceous Particles: An Investigative Review", Aerosol Science and Technology) is included

Page 4, Line 169: What type of scaling was used? Reference?

Response: We have now specified this in the text, and provided references; "Hygroscopic growth of SO4 was included, scaling with relative humidity according to Fitzgerald (1975). See Myhre et al. (2004) for a discussion on the impacts of this choice."

Page 5, Line 175: I would say that it would be useful to give a brief summary of the impact of using a single model's kernel in this manuscript too, given how central AFEs are for the results presented here. No need to be long – a few sentences would suffice.

Response: See the revised section above

Page 5, Line 177: I presume by "resulting" it is meant "modelled"? If so, please change, clearly mentioning that this is from the new HTAP2 simulations.

Response: The sentence now reads "The direct RF from a given aerosol species due to a 20 % emission reduction was then estimated by multiplying the modelled aerosol burden change profile ?BD (rom a given HTAP2 model and experiment) with the OsloCTM2 AFE distribution for that species and point in space and time (month of the year).", which is hopefully more clear.

Page 5, Line 178: Please remind the reader that "time" here corresponds to the month of the year.

Response: This information is now added.

Page 6, Lines 214-215: Suggest rephrasing the part of the sentence after the second

comma with "which gives an estimate of inter-continental transport in two dimensions (ignoring the vertical)".

Response: We agree that this improves the sentence and have followed the reviewer's suggestion.

Page 8, Lines 301-312: This is a peculiar feature and needs some further explanation. It is a bit too hand-wavy to say that nudging may be responsible. If so, why would it mainly appear in SO4 and OA, but not BC? Maybe it has to do with effects of aerosol emission reductions on oxidants in the models? SO4 and OA would be affected by this, but not BC.

Response: We appricate the useful suggestion and have included the following sentences emphasizing oxidant changes as a potential cause of the concentrations changes: "Regional increases in aerosol concentrations imposed by emission reductions can be observed for SPRINTARS and CAMchem, and to a smaller extent also for the CHASER models, GEOS5 and C-IFS (not shown, but visible in the globally averaged RBUreduced and MDEreduced plots for OA in Fig. 4). This occurs mainly for OA and SO4. Conceivably, aerosol emission reductions may in these models be influencing the level of oxidants, which would have feedbacks on the concentrations of OA and SO4. A model study by Shindell et al. (2009) demonstrates the importance of aerosol-gas interactions to the climate impact of mitigations. They point out that the effect on oxidant changes on SO4 concentrations are stronger in oxidant-limited regions with high SO2 emissions, and that greater parts of the industrialized Northern Hemisphere is, in fact, oxidant limited (Berglen et al., 2004)"

Page 8, Lines 315-319: Is this in agreement with what other studies that examined long-range transport of pollution to the Arctic (e.g. Shindell et al., 2008) have found? Worth mentioning.

Response: Thank you for this comment; consistency with previous studies is indeed relevant to mention. We do see some similarities between the Shindell et al. paper

and ours: high-altitude changes in pollution levels in the Arctic tended to originate from East and South Asia, while low-level changes were dominated by changes in Europe. In addition we see a strong influence on the Artic from Russia, which is also seen in other studies (Sand et al., 2013b; Stohl, 2006). We have included a few sentences on this at the given location in the text.

Figure 6: I suggest reminding the reader in the caption that these are inferred from one model, and which model this is.

Response: This information is now added to the caption.

Page 9, Line 346: I suggest clarifying that panel (a) is for BC. This is more important than mentioning the panel.

Response: We agree, and have now rephrased the sentence to clarify this.

Page 9, Lines 354-356: Worth mentioning that the vertical increase in the Middle East is not as steep, presumably due to the lower occurrence of clouds in this area.

Response: Absolutely, we have now included a sentence on this.

Page 10, Lines 387-388: Could it also be the lower insolation in this region?

Response: Lower insolation could at least be a contributing cause for the lower AFE values in Russia, and we now include this in our suggested explanation.

Page 10, Line 402: Worth citing the recent paper by Kasoar et al. (2016) here, as it also discusses thoroughly the causes of diversity in three different models when it comes to simulating climate impacts of identical regional SO2 emission perturbations.

Response: The paper, which is also part of the ACP special issue on HTAP, should indeed be cited, and is now included among the other references.

Table 4: I may be missing something, but shouldn't the values in Table 4 be consistent with the values in Figure 10? I cannot see this fully being true: For example, for the

first bar in Fig. 10 - representing North America - the domestic influence seems to be responsible for much more than half of the SO4 forcing, but then in Table 4 it appears as if it is just above 60%. Please check consistency (for the whole table) and/or explain.

Response: This is well spotted, and we understand that the differences look incoherent. The reason that the numbers are not equal is that for the RERER calculations, numbers (e.g., domestic contribution to the RF) are calculated relative to the experiment where global emissions were reduced by 20%. Conversely, in Figure 10, the corresponding numbers are calculated relative to the sum of the given region's forcing caused by all the six major experiments. As the summed RF following emission reductions in our six source regions is not quite as large as a 20% emission reduction all over the world, the RERER numbers in Table 4 are relative to larger numbers and will therefore be smaller than the corresponding numbers of Figure 10. We have tried to express this in the text.

Table 4: I suggest mentioning what the range indicated next to the means represents.

Response: That information should definitely be there – this is now fixed.

Page 13, Line 520: Please add "surface" before "albedo".

Response: The word is now added.

Page 13, Lines 526-530: As mentioned in the abstract as well, I suggest that you clearly state at the beginning of this paragraph that in most cases the local influence is the dominant, though with important contributions from remote regions. And then proceed with outlining the cases where remote is stronger than local (as is done already).

Response: We have added an extra couple of sentences here to stress this.

Page 13, Lines 531-533: In the first sentence of this paragraph, it is mentioned that the effect of "vertically resolved concentrations" is examined, while in the next sentence it is stated that "Using vertically resolved AFE distributions strengthens. . .". Which of the two is examined, the influence of smoothed concentrations or of smoothed AFEs?

Earlier it is mentioned that the effect of both is examined. Please clarify.

Response: This was indeed unclear; we have now rephrased this section to clarify.